# Resilient water infrastructure partnerships in institutionally complex systems face challenging supply and financial risk tradeoffs

A. L. Hamilton [1,2] ✉, P. M. Reed [1], R. S. Gupta[1], H. B. Zeff[3,4] & G. W. Characklis [3,4]

As regions around the world invest billions in new infrastructure to overcome increasing water scarcity, better guidance is needed to facilitate cooperative planning and investment in institutionally complex and interconnected water supply systems. This work combines detailed water resource system ensemble modeling with multiobjective intelligent search to explore infrastructure investment partnership design in the context of ongoing canal rehabilitation and groundwater banking in California. Here we demonstrate that severe tradeoffs can emerge between conflicting goals related to water supply deliveries, partnership size, and the underlying financial risks associated with cooperative infrastructure investments. We show how hydroclimatic variability and institutional complexity can create significant uncertainty in realized water supply benefits and heterogeneity in partners' financial risks that threaten infrastructure investment partnership viability. We demonstrate how multiobjective intelligent search can design partnerships with substantially higher water supply benefits and a fraction of the financial risk compared to status quo planning processes. This work has important implications globally for efforts to use cooperative infrastructure investments to enhance the climate resilience and financial stability of water supply systems.

Water scarcity impacts are increasingly frequent and severe in many parts of the world due to climate change, economic growth, and regulatory change[1-4]. Many regions are investing heavily in new infrastructure such as reservoirs, canals, and groundwater recharge facilities to bolster their water supply reliability and avoid negative impacts on public health, food security, power supply, and ecosystems[5]. In the U.S., federal and state funding for water systems has declined in recent decades, so that the large majority of water infrastructure spending comes from the local level[6,7]. Much of this

funding comes from municipal bonds issued by local governments and water utilities, which provide up-front capital for projects in return for the principle plus interest to be repaid over many years. When investing in new infrastructure, water providers must be careful to balance the benefits of supply investments with their associated long-term debt obligations that must be repaid using water sales revenue[8,9]. This financial risk is especially important in light of concerns over the creditworthiness of drought-exposed water providers, concerns that can lead to higher borrowing rates

[1]School of Civil and Environmental Engineering, Cornell University, Ithaca, NY, USA. [2]Confluency, Chicago, IL, USA. [3]Department of Environmental Sciences and Engineering, Gillings School of Global Public Health, University of North Carolina at Chapel Hill, Chapel Hill, NC, USA. [4]Center on Financial Risk in Environmental Systems, Gillings School of Global Public Health, UNC Institute for the Environment, University of North Carolina at Chapel Hill, Chapel Hill, NC, USA. ✉e-mail: andrew.hamilton.water@gmail.com

and contribute to growing water affordability challenges in many regions[10-14].

Collaborative partnerships, water supply regionalization, and consolidation have garnered attention for facilitating investment in large-scale infrastructure. Multiple water providers collaborating to finance, build, and operate shared facilities with joint benefits can be more cost-effective than independent provisioning due to economies of scale, reduced redundancy, improved access to capital, and diversification of supply and demand patterns[15-20]. Regions across the U.S., from California to Kentucky, have encouraged cooperative regional infrastructure investments with funding programs and technical assistance[21,22].

However, collaborative infrastructure investments also carry risks. Modern water supply systems are governed by a complex set of hydrologic, infrastructural, and institutional drivers and uncertainties[23-25]. This presents a significant modeling challenge for water providers trying to estimate the expected benefits and costs of expensive infrastructure investments. For example, understanding the value of groundwater recharge facilities requires detailed modeling of short-timescale dynamics (e.g., atmospheric rivers or snowmelt-driven flood events), long-timescale trends (e.g., climate change and aquifer depletion), and time-evolving infrastructure constraints, water rights, and regulations[26-29]. Moreover, water supply systems in many regions are increasingly interconnected, which can lead to complex emergent behaviors within the coupled network because each water providers' future supply reliability depends not only on its own actions and capabilities, but also on the actions and capabilities of other water providers in the network[30-32]. Shared water supply facilities that reduce system redundancy can also increase vulnerability during extreme conditions. Collaboration amplifies these challenges because it requires understanding not only the total aggregate benefits and costs of infrastructure investment, but also the uncertain distribution of these benefits and costs to individual partners over time[33-35].

To date, little work has been done to help water providers understand the local-scale benefits and risks of investing in regional infrastructure partnerships. Water supply planning models are often highly aggregated, with many water providers grouped together, which limits their ability to discover local-scale tradeoffs for individual partners[23,33]. Moreover, little guidance exists to facilitate the design of resilient and equitable water supply infrastructure partnerships in institutionally complex systems. Existing economic planning frameworks for partnership design and cost apportionment are generally based on expected value benefit-cost analysis and/or game theoretic methods[36-40]. These methods are not well suited to realistic systems characterized by a large number of potential partners, interactive network effects, high uncertainty, and multiple conflicting objectives. In practice, infrastructure partnerships tend to emerge through pre-existing relationships or ad hoc arrangements. However, as we show in this study, this can lead to significant regret in complex water systems where partnership performance is challenging to predict.

In this work, we highlight the substantial water supply tradeoffs and financial risks that must be navigated by water providers when designing regional infrastructure partnerships. To do this, we develop an approach that uses detailed ensemble modeling of an interconnected water supply network under uncertainty, combined with a multiobjective intelligent search to discover the optimal tradeoffs in infrastructure partnership design (Supplementary Fig. S1). We apply this approach in the context of ongoing canal expansion and groundwater bank development efforts in the San Joaquin Valley region of California, representative of the state's efforts to encourage innovative partnerships through its Water Resilience Portfolio Initiative[21]. California is poised to spend billions of dollars on collaborative water infrastructure investments in the coming years[7,41-43]. We find a wide diversity of alternative infrastructure partnerships that exhibit strong tradeoffs between the size of a partnership and partner-level financial risk, and between water supply benefits for the investing partners and the rest of the region. Our findings also emphasize the importance of accounting for the substantial uncertainty in infrastructure performance that emerges due to California's extreme hydroclimatic variability. Uncertainty in overall infrastructure performance is accompanied by high degrees of heterogeneity in the water supply benefits and financial risks experienced by individual water providers under different partnerships. Lastly, we demonstrate the advantages of our multiobjective intelligent search framework through a baseline comparison focused on the ongoing efforts to rehabilitate the Friant-Kern Canal. As an example, one of the candidate partnership designs discovered using our intelligent search procedure achieves 58% higher water supply gains while also significantly lowering the risk of extreme cost burdens for investing water providers when compared to the existing status quo canal expansion partnership. These insights are only attainable by moving beyond traditional ad hoc planning practices, expected value benefit-cost frameworks that do not resolve key differences in partners' individual water supply and risk tradeoffs, and highly aggregated regional water supply models that fail to resolve the institutional complexities and local-scale dynamics that drive consequential differences in partners' investment benefits and risks. Our results have important policy implications for California and broader insights for other regions working to develop collaborative investment partnerships to increase the resilience of their water supply systems.

## Results
### Water supply infrastructure partnership design
The San Joaquin Valley region in California is home to over four million people and over five million acres of irrigated farmland[44]. The region's long-standing water supply challenges have been amplified in recent years as climate change has made drought more frequent and severe, and as chronic groundwater overdraft has caused dry wells, widespread subsidence, water quality issues, and restrictive new regulations[21,44-46]. Local and state agencies are working to address these challenges by investing in new infrastructure and cooperative water management strategies[20,26,42,47]. In this study, we analyze two key ongoing infrastructure initiatives in the Tulare Lake Basin region of the San Joaquin Valley (Fig. 1). First, the Friant-Kern Canal is an important structure that primarily conveys San Joaquin River water south from Millerton Lake as part of the federal Central Valley Project. However, the capacity of the canal has been reduced by up to 60% due to subsidence-related damage[48,49]. In 2021, a collection of water providers known as the Friant Contractors decided to invest ~$50 million to rehabilitate the canal and expand its capacity, with various federal and state programs providing support for the rest of the estimated $500 million cost[50]. We consider only the $50 million borne by local users in this study. The fact that we find significant financial risks even with this unusually favorable cost structure serves as a warning for other less-subsidized infrastructure investments in the region. We also consider a second infrastructure investment in a new groundwater bank, which would route surplus water available in wet periods into infiltration basins to replenish the aquifer and allow for additional groundwater pumping during dry periods[20,27,28,42]. We estimate this project would also cost local water providers $50 million based on cost estimates for other groundwater storage projects being developed throughout the region[43].

Using the cooperative infrastructure investment context of the Friant-Kern canal rehabilitation and its potential augmentation with a new groundwater bank, we simulate the performance of alternative infrastructure partnerships using the California Food-Energy-Water System (CALFEWS), a highly detailed water resource systems model that simulates daily reservoir operations, environmental flow rules, water rights, interbasin transfer projects, and conjunctive surface and groundwater supply management at the level of individual water

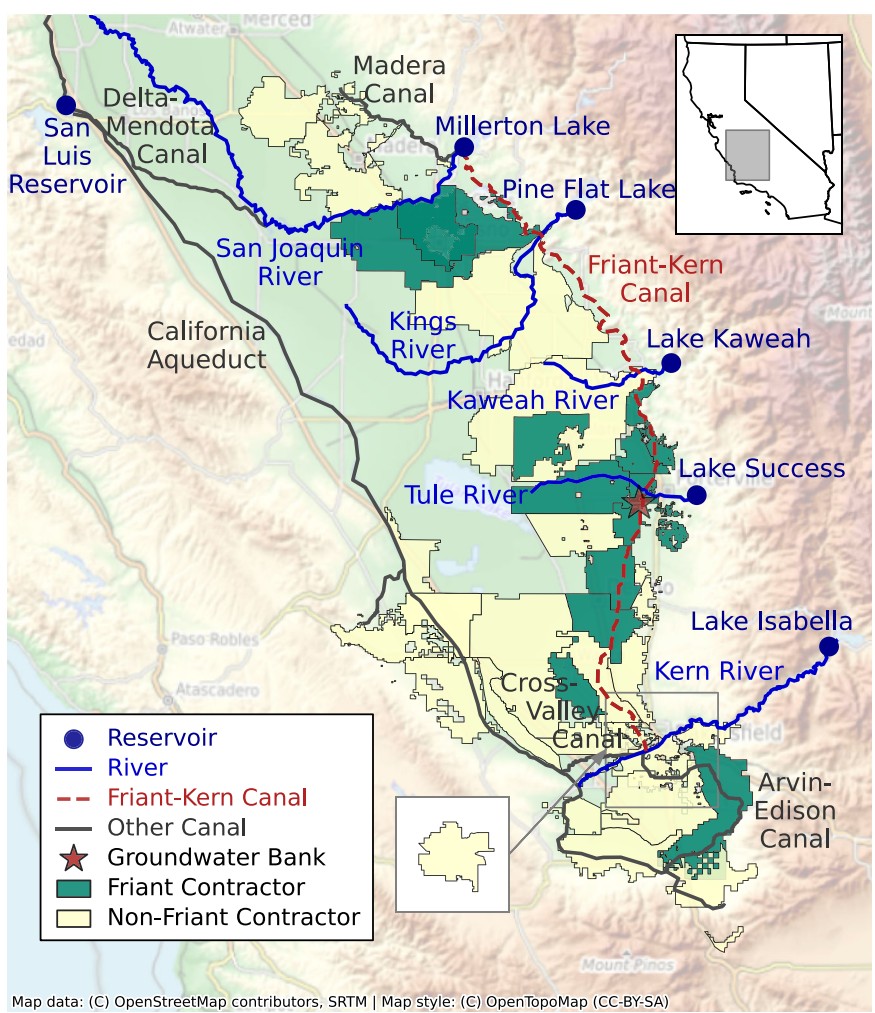

**Fig. 1 | Map of the study area, showing major reservoirs, rivers, canals, and water providers of the Tulare Lake Basin region in California's Central Valley.** The infrastructure projects considered in this study are an expansion of the Friant-Kern Canal (FKC, dotted red line) and a new groundwater bank near the FKC and the Tule River (red star). "Friant Contractors" are water providers with contracts to receive Central Valley Project water from the FKC. "Non-Friant Contractors" are other water providers in the region without Friant contracts. The bottom center inset shows KCWA Improvement District No. 4, a water provider that overlaps with other providers in the square-designated region of the main map.

Map data: (C) OpenStreetMap contributors, SRTM | Map style: (C) OpenTopoMap (CC-BY-SA)

providers[23,33]. The high-fidelity system representation of the model allows us to generate new insights into infrastructure partnership design compared to existing lower-fidelity water supply planning models that cannot resolve the multi-timescale dynamics of managed aquifer recharge or the multi-spatial scale distribution of water supply benefits and financial risks for individual water providers[23,33]. Each partnership is evaluated across an ensemble of daily time step 30-year hydrologic sequences from a synthetic streamflow generator that captures the spatial correlation across the state as well as the inter-annual persistence of wet and dry periods (see Methods and Supplemental Note S1). This allows us to explore the impact of California's severe internal hydroclimatic variability on the water supply benefits and financial risks resulting from these infrastructure investments[51,52]. The synthetic streamflows used in this study serve two major technical benefits: (1) they permit a well-founded statistical representation of the plausible flood-drought regimes that regional infrastructure invest-ments could potentially face in the near term, and (2) they are inten-tionally strongly optimistic in neglecting longer-term climate changes to show that even without climate change effects, there are immense uncertainties that challenge our understanding of investment part-nerships' water supply and financial risk tradeoffs. Finally, for an additional check on partnership robustness, we compare the perfor-mance of two example partnerships under 20 alternative streamflow

scenarios for 2021–2050 from an existing dataset generated using a multi-model ensemble of downscaled climate model runs for Cali-fornia from the Climate Model Intercomparison Project Phase 5 (CMIP5) and the Variable Infiltration Capacity (VIC) model (See Sup-plementary Note S1 for details)[53–56].

Partnership performance is assessed over the 30-year simulation (a common municipal bond payback period[57]) according to four decision-relevant metrics related to partnership size, water supply benefits, and financial risk. We combine this ensemble modeling framework with a multiobjective intelligent search framework that leverages the Borg MultiObjective Evolutionary Algorithm (MOEA)[58,59] to test ~300,000 candidate infrastructure investment partnerships to discover the best-performing partnerships that represent the optimal tradeoffs across the four metrics (Supple-mentary Fig. S1). The elements of partnership design that are opti-mized can be described with three questions: (1) In which project does the partnership invest (canal expansion, groundwater bank, or both)? (2) Which subset of 40 water providers in the region should participate? (3) What "ownership share" should be assigned to each partner, governing its capacity in the project and its share of annual debt payment obligations? Additional details on our partnership design and evaluation framework can be found in Methods and Supplementary Notes S2 and 3.

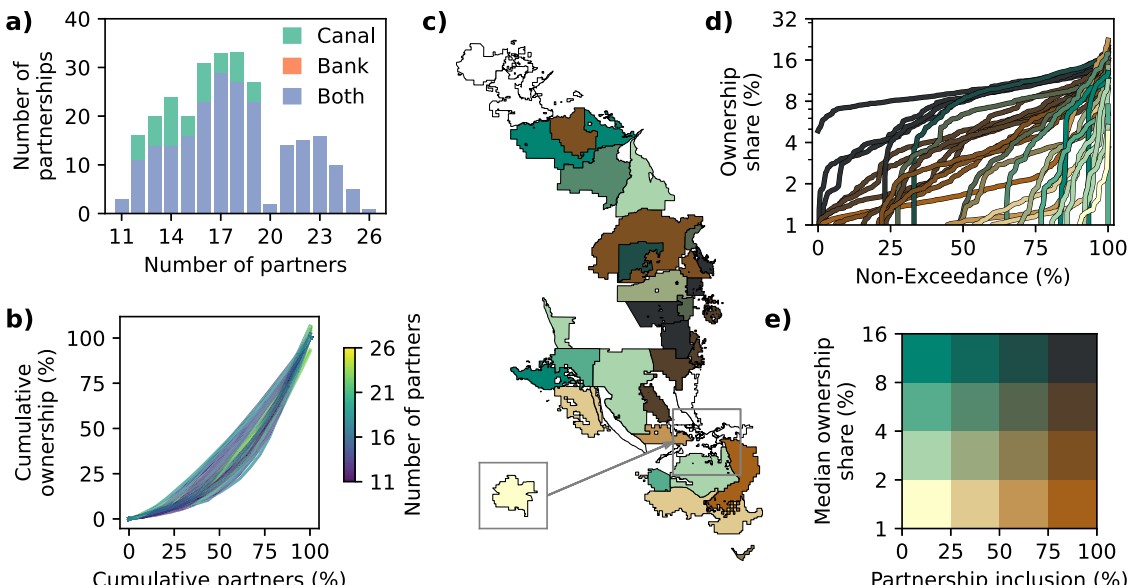

**Fig. 2 | Variation in partnership design across the optimal tradeoff set.**
**a** Distribution of partnership sizes, colored by the type of infrastructure.
**b** Concentration of ownership shares in each partnership, colored by the size of each partnership. **c** Map of water providers, colored according to the bivariate palette in **e**. Water providers that do not participate in any optimal tradeoff partnerships are colored white. **d** Non-exceedance curves showing the distribution of ownership shares (log scale) across the optimal tradeoff set for each water provider, colored according to the bivariate palette in **e**. **e** Bivariate color palette to classify each provider based on the percentage of optimal tradeoff partnerships in which it participates and its median ownership share within those partnerships.

## Understanding diverse partnership design options

Our multiobjective intelligent search process yields a set of 270 unique partnership structures which are highly diverse in terms of their design (i.e., different combinations of partner districts and their levels of relative investment) as well as their performance across different objectives—this broad suite of alternatives collectively represents a wealth of information which would not have been available using traditional ad hoc partnership planning processes which typically consider only a small number of candidate arrangements. Each candidate partnership within the set of 270 approximately Pareto-optimal partnerships (hereafter referred to as "optimal tradeoff partnerships") represents a different balance of compromises across the four conflicting objectives.

The majority of optimal tradeoff partnerships (83%) are found to invest in both canal expansion and groundwater banking (Fig. 2a). This suggests a synergistic relationship. The surface water gains from the Friant-Kern canal expansion generally occur during relatively high-flow periods, when available water supply exceeds immediate demand, so it is complementary to pair the expansion with additional storage[29,60]. The remaining 17% of optimal tradeoff partnerships invest only in canal expansion, with no partnerships investing in groundwater banking alone. This highlights the importance of considering the interactive effects across alternative infrastructure projects and the potential for synergistic gains when bundling regional investments together.

There is a high degree of diversity in the way that partnerships can be designed. The optimal tradeoff partnerships range in size from 11 to 24 water providers, with 16–19 being the most common partnership sizes (Fig. 2a). The fact that no optimal tradeoff partnership contains fewer than 11 partners supports the vision of California's Water Resilience Portfolio Initiative, which touts the benefits of collaboration for efficiently meeting the state's water supply goals[21]. Larger partnerships are expected to have larger and more diverse sources of water supply, water demand, and local infrastructure and, thus, are more capable of fully utilizing the new shared infrastructure throughout the year and across a range of conditions.

There is also diversity in the concentration of ownership across the partnerships. There are no partnerships that distribute ownership equally (e.g., 20 partners with a 5% share each). Instead, most partnerships have a few partners with disproportionately large ownership shares and a larger number of partners with rather small shares, with the magnitude of these differences varying across the optimal tradeoff set (Fig. 2b).

Considering the geographic context of these partnerships (Fig. 2c–e), we find that certain providers almost always participate and generally carry large ownership shares (dark gray), while others rarely participate and tend to take rather small shares (yellow/light green/light brown). There are also providers that regularly participate with small shares (medium brown) or that irregularly participate with large shares (teal). The set of providers that participate in over half of the optimal tradeoff partnerships and that have a median ownership share above 4% (top right quadrant of Fig. 2e) are heavily concentrated in the central region along the Friant-Kern Canal in the vicinity of the Tule River (Figs. 1 and 2c). These are the providers most heavily impacted by the heavy subsidence in this area and the subsequent restriction in canal conveyance capacity. Many of these providers are Friant Contractors, but there are also several Friant Contractors that participate infrequently and/or with small shares, as well as several non-Friant districts that can play an important role in investment partnerships, as we will see in subsequent sections.

These results demonstrate the complexity of partnership design and the range of roles that different water providers can play in partnership creation given their unique contexts: the local infrastructure networks, water rights, hydrologic context, and other factors that impact their ability to procure and store additional surface water as a result of the collaborative investment. Moreover, many providers can span a range of ownership shares across the optimal tradeoff set depending on how the rest of the partnership is constructed (Fig. 2d). This results from a large number of provider interactions across the coupled statewide-to-local-scale infrastructure network (e.g., shared canal space) as well as interactions in the institutional space (e.g., water contracts and groundwater banking arrangements). These interactions can lead to path-dependent system dynamics across the region, where each providers' water supply operations can impact the availability of water and infrastructure capacity for the other providers in

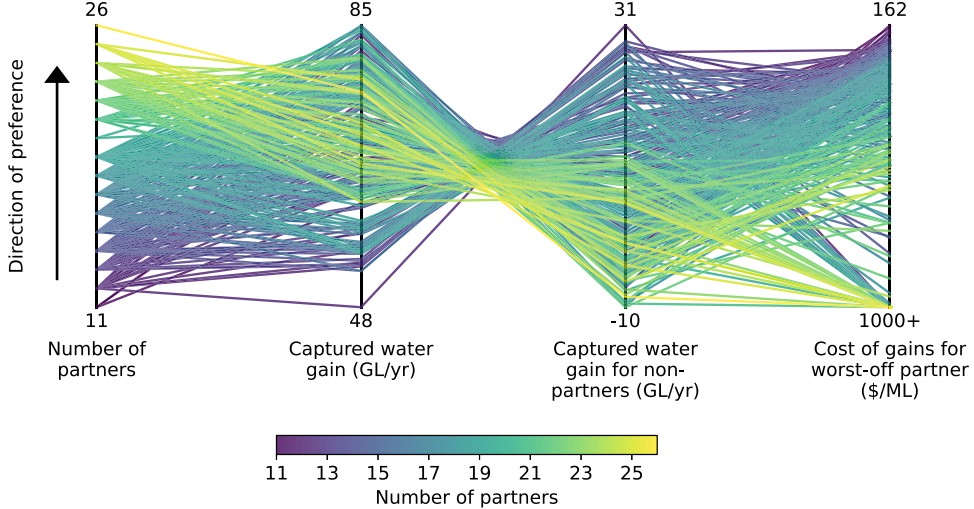

**Fig. 3 | Parallel coordinate plot showing the performance of different infra-structure partnerships relative to four decision-relevant objectives (vertical axes).** Each colored line represents one of the 374 optimal tradeoff partnerships. The location where a partnership intersects each vertical axis represents its performance with respect to that objective. Each objective is oriented such that the preferred direction is up (e.g., maximum water gain, minimum cost). Thus, ideal performance would be a horizontal line across the top, while diagonal segments represent tradeoffs between conflicting objectives.

the network. This highlights the importance of capturing local-scale dynamics and network interactions between water providers when designing collaborative infrastructure investments[23,33].

## Key performance tradeoffs for water supply investment partnerships

We find that water providers face severe tradeoffs across four decision-relevant metrics when designing collaborative infrastructure invest-ments (Fig. 3). The optimal tradeoff partnerships can increase average surface water deliveries to partners by anywhere from 48 to 85 GL/year. For context, 85 GL (69 kAF) is equivalent to 13% of Millerton Lake's capacity, or 3.4% of the average annual groundwater overdraft for the entire San Joaquin Valley[44]. This represents a meaningful increase in surface water deliveries at the local scale, which could help partners reduce their over-reliance on groundwater pumping in order to meet their obligations under the Sustainable Groundwater Man-agement Act, although this would need to be only one piece in a larger portfolio of water supply resiliency investments to address the scale of the challenge of groundwater overdraft in the Valley[42].

In addition to water supply volumes, we also calculate the worst-off partner cost of gains, defined as the annual debt service payment divided by the annual captured water gain for the worst-off partner in each partnership. The 90th percentile cost of gains across alternative hydrologic scenarios, which we use as a proxy for partner-level financial risk, ranges from $162/ML to over $1000/ML in different partnerships (Fig. 3; see Methods for the detailed definition). For context, water providers in this region generally charge $32-154/ML ($40–190/AF) for irrigation water[61], which is used to pay for debt ser-vice on infrastructure investments, as well as the cost of procuring water and other operating expenses. Financial distress can occur when a water provider agrees to future debt obligations as part of an infra-structure partnership but does not end up receiving substantial water supply benefits from the project. Optimal tradeoff partnerships with very high costs of gains may be unable to raise customer rates high enough to pay off the debt associated with the new infrastructure. This could lead to default, credit downgrade, deferred maintenance, or other challenges[9–11]. Significant increases in customer water rates are also problematic due to the water affordability and water quality challenges in many low-income communities, as well as the thin profit margins in many agricultural regions where higher rates for irrigation water can make farming infeasible[12,14,44,62,63].

One strong tradeoff that emerges from these results is that the largest partnerships tend to produce the largest financial risks for their worst-off partners. For example, over half of partnerships with 24+ partners have at least one partner paying over $1000/ML, while each partnership in which all partners pay under $200/ML has 16 or fewer partners (Supplementary Figs. S2 and S3). It is intuitive that beyond a certain point, a larger number of partners can make it harder to satisfy all partners' needs. Yet this finding also points to an important tension in California's recent efforts to incentivize large-scale collaborative water supply investments[21]. Large-scale collaboration in infrastructure investment can indeed be beneficial up to a point, as evidenced by the fact that all optimal tradeoff partnerships have 11+ partners (Fig. 3). However, it is also critical to understand the limits of large-scale col-laboration and the significant financial risks that can arise from these arrangements, in order to ensure that all partners are receiving ade-quate benefits to justify their debt. There are also likely additional tradeoffs beyond those quantified in our work, such as costs, delays, and other challenges related to coordinating and contracting across a large and diverse set of partners with their own incentives and his-torical relationships[16].

We also find a strong tradeoff between water supply benefits for the partners vs. other non-partner water providers in the region (Fig. 3). New infrastructure projects alter water providers' path-dependent supply and demand behaviors, with ripple effects throughout the broader interconnected water supply network. This can lead to indirect effects on water availability for other water providers in the region that are not party to the investment. We find that the non-partner impacts vary greatly across the optimal tradeoff set, from gains of 31 GL/year to losses of 10 GL/year on average. In general, the partnerships with the largest captured water gains for the partners tend to significantly reduce deliveries to non-partners (Supplementary Fig. S4). This sug-gests that these infrastructure partnerships benefit not only from newly captured "surplus" water that would have otherwise flowed out of the region but also from reallocated water that would have otherwise been delivered to their neighbors.

This raises important questions about the extent to which non-participating water providers have the ability to block infrastructure investments that could negatively impact them. Water providers investing in new infrastructure in California must navigate the state's complex web of different water rights, environmental laws, and administrative procedures[25,26]. In particular, as in other western states,

many changes related to water rights and diversion and storage patterns require that the change will cause "no injury" to other water rights holders[64]. Third parties can object if they believe any injury has occurred, and outcomes are adjudicated in court or with the State Water Resources Control Board on a case-by-case basis based on the unique hydrologic and legal context. Paying for hydrologic modeling studies and legal analysis to support the validity of any changes to water diversion or storage patterns, as well as potentially compensating any injured parties, can add significant expense and delays to the approval process[65]. For example, previous studies have found that transaction costs related to hydrologic modeling and legal support required for approval of water rights transfers in California and Colorado can increase the total cost of the transfer by 100% or more[66,67]. In California, the diversity of legal regimes and water laws in the state (e.g., prior appropriative rights vs. riparian rights vs. federal/state water supply contracts) further complicates the evaluation of third-party injury related to alternative water supply options. More guidance is needed from the state to streamline procedures and facilitate more collaborative partnerships. When we exclude partnership designs from our optimal tradeoff set, which are expected to reduce total average water deliveries to non-partners to account for potential political and legal constraints, the best achievable captured water gains for partners are reduced by 6% (Supplementary Fig. S5). Excluding partnerships that would cause injury to individual providers (rather than in aggregate) or in individual years or hydrologic scenarios (instead of expected value across many scenarios/years) would be expected to substantially reduce the set of feasible partnerships further. This negative interaction represents a major challenge for supply reliability and groundwater sustainability efforts and points to the need for more coordinated regional infrastructure planning efforts to develop synergistic water supply portfolios with regional benefits and minimal third-party injuries, as well as programs to mitigate and compensate for these injuries.

It is perhaps surprising that such a wide diversity of performance tradeoffs is possible, given that each of these 270 optimal tradeoff partnerships invests in the exact same canal expansion project, sometimes bundled with a groundwater bank (Fig. 2a). This demonstrates the critical and underappreciated role of partnership design itself in governing the performance of collaborative infrastructure investments. The variety of partnership designs associated with different performance tradeoffs means that it is not possible to establish a single "best" partnership a priori without first establishing decision-making preferences (i.e., how much to weigh partnership size vs. captured water gains vs. regional non-partner impacts vs. worst-off partner cost). Moreover, the striking performance tradeoffs that emerge from nuanced changes in partner selection and ownership distribution highlight the significant advantages of pairing detailed water supply models like CALFEWS[23] (capable of resolving daily-timescale coupled hydrologic, infrastructural, and institutional dynamics at the scale of individual water providers) with multi-objective intelligent search algorithms like the Borg MOEA (capable of exploring a much wider range of candidate partnership designs compared to traditional ad hoc planning processes).

## Navigating uncertainty and heterogeneity in tradeoffs

We also find that uncertainty from hydroclimatic variability can have significant impacts on water supply and financial risk consequences for infrastructure investments, and thus uncertainty should be systematically analyzed in partnership design. When an investment partnership borrows money in municipal bond markets to pay for a new infrastructure project, it is subject to significant uncertainty related to the future realized water supply benefits and sales revenues. This uncertainty can be highly decision-relevant when there is a wide range of performance values across plausible alternative future scenarios. The metrics in Fig. 3 represent aggregate performance computed

across an ensemble of 79 sampled 30-year sequences of multi-site correlated daily streamflows. This ensemble captures a wide but plausible range of hydrologic conditions that investments could confront over a 30-year bond payback period (see Methods and Supplementary Notes S1, S2). The captured water gain metric in Fig. 3 is calculated as the mean values across the 79 sampled 30-year daily streamflow sequences, while the worst-partner cost of gains metric is calculated using the 90th percentile across scenarios. However, a deeper examination of performance in individual sampled 30-year daily hydrologic sequences reveals significant levels of uncertainty in the benefits that water providers can expect due to internal hydroclimatic variability. These uncertainties can be highly meaningful in the context of risky large-scale infrastructure investments.

We now highlight a "Compromise Partnership" selected as an example for navigating the tradeoffs in the optimal partnership set. The Compromise Partnership has 16 partners investing in both canal expansion and groundwater banking (Fig. 4a) and is selected for its relatively high performance across all four performance objectives (see Methods). Like all partnerships in the optimal tradeoff set, the Compromise Partnership is subject to significant performance uncertainty (Fig. 4b). Although the expected value of captured water gain is 76 GL/year, the captured water gain in individual 30-year daily hydrologic sequences can range from 43 to 109 GL/year, or 57% to 143% of the expected value. The captured water gain for non-partners is even more variable. Although the partnership leads to a minor reduction in non-partner deliveries to on average (0.5 GL/year), the impact in individual hydrologic scenarios spans a wide range, from a reduction of 81 GL/year to a gain of 26 ML/year. The worst-partner cost of gains also spans a wide range, from $105/ML in the best scenario (58% of the 90th percentile metric of $180/ML) to $407/ML in the worst scenario (227% of the 90th percentile metric). This cost differential could easily be the difference between a project that affordably improves surface water access and one that provides little water supply benefit and overburdens water providers with debt and their customers with rate increases.

The benefits and risks of different partnerships can be very unevenly distributed across the project partners due to the complexity of the interconnected water supply system dynamics and the heterogeneity of local contexts for water providers. The partners within the Compromise Partnership experience a range of expected costs of gains for their water supply benefits (Fig. 4c), which is common across the optimal tradeoff partnerships. The heterogeneity of expected costs stems from the similarly heterogeneous expected captured water gains at the partner level (Supplementary Fig. S8a). The latter is not inherently problematic so long as each partner's captured water gain is appropriately matched to its ownership share in the project and, thus, its share of the annual debt payments. For example, if Provider A receives twice as much captured water gain as Provider B, but also makes annual debt payments that are twice as large, then the two providers are effectively paying the same unit cost for their captured water gains. However, we find the ownership shares to be imperfectly matched in this and other partnerships, so that some providers pay more than their "fair share" on average, and others pay less (Fig. 4c).

Moreover, partners can experience widely varying degrees of uncertainty related to their individual performance tradeoffs (Supplementary Fig. S8). The aggregate cost of gains for the Compromise Partnership (Fig. 4c, black distribution) has a relatively low chance of exceeding $100/ML in any of the sampled hydrologic scenarios. However, the cost of gains for the worst-off partner (red distribution) has a much wider range, reaching over $400/ML in the most challenging scenario. This risk is not evenly distributed, with some partners experiencing a disproportionate share of extreme costs (e.g., three out of sixteen partners experiencing worst-case costs over $300/ML) and others experiencing uniformly low costs across the sampled hydrologic scenarios (e.g., four partners experiencing worst-case costs

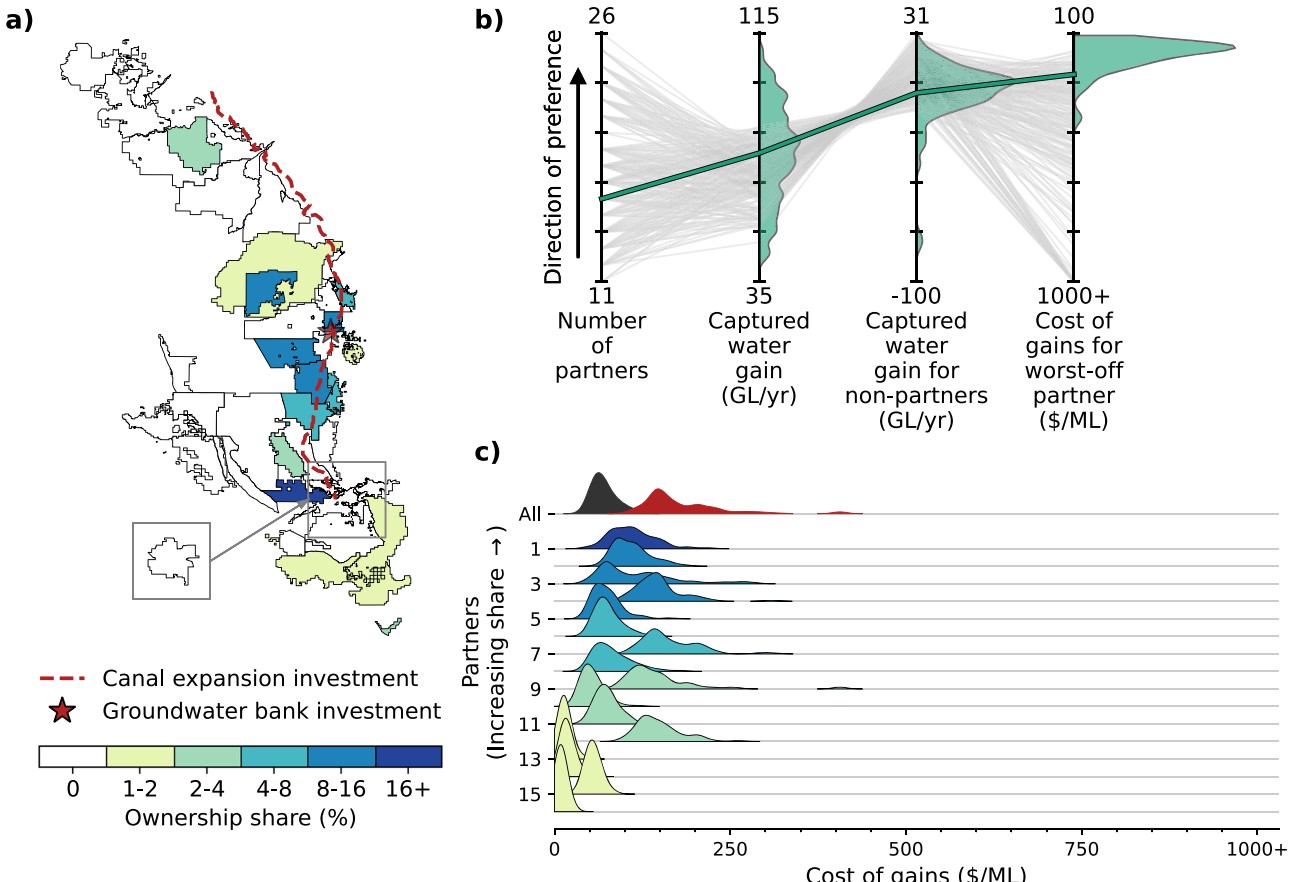

**Fig. 4 | Uncertain and heterogeneous performance of the Compromise Partnership. a** Map showing participating water providers for the Compromise Partnership, which invests in both canal expansion and groundwater banking. Partners are colored by their ownership share in the partnership, while non-partners are colored white. **b** Parallel coordinate plot showing partnership performance across four conflicting objectives. Each gray line represents the performance of a different optimal tradeoff partnership, aggregated across the 79 sampled 30-year daily hydrologic sequences. The green line represents the aggregated performance of the Compromise Partnership, while the green shaded areas show the probability distribution of single-scenario performances for the Compromise Partnership. **c** Heterogeneity of partner-level performance for the Compromise Partnership. Each shaded area labeled 1–16 represents the distribution of costs of gains for a different water provider partner across the 79 sampled 30-year daily hydrologic sequences. The partners are sorted and colored by ownership share. The distributions labeled "All" represent the partnership-level aggregated costs (black) and the costs for the worst-off partner across alternative scenarios.

under \$100/ML). This is critical because the heterogeneity of water supply benefits and financial risks could threaten the cooperative stability of the partnership itself if partners do not perceive the investment to be sufficiently fair and locally beneficial[30,33].

As noted before, these results (Fig. 4) are a strongly optimistic portrayal of partnership performance under conditions of uncertainty and heterogeneity, which reinforces the importance of our results given the impacts of anthropogenic climate change. Climate change is expected to make dry conditions increasingly frequent and severe in California and many other regions[2,68,69]. Our expressly optimistic framing of future hydroclimatic conditions in California neglects any nonstationary effects from anthropogenic warming or low-frequency decadal climate oscillations. However, even under such optimistic conditions, the extreme levels of hydroclimatic internal variability experienced in California[51,52] lead to significant decision-relevant uncertainty in cooperative infrastructure investment outcomes over the next several decades. Water providers, therefore, must find a way to improve surface water reliability and reduce groundwater over-drafts without exacerbating the growing water affordability concerns for agricultural and urban water users[12–14]. We caution that decision-makers should carefully consider the impacts of uncertainty and financial risk before committing to significant debt associated with new infrastructure partnerships. However, this will require updated planning frameworks that move beyond expected value-based benefit-

cost analyses (e.g., exploratory modeling and robust decision-making approaches[70–74]), as well as detailed water supply models that can account for complex local-to-regional scale dynamics in coupled hydrologic, infrastructural, and institutional systems under diverse conditions[23]. Flexible planning frameworks such as Engineering Options Analysis, Dynamic Adaptive Policy Pathways, and adaptive contract structures can also help decision-makers to design for flexibility and delay expensive irreversible investment decisions until they have gathered more information about likely future conditions[75–79].

## Regret of the current baseline partnership design

Lastly, we highlight the potential to meaningfully improve existing ad hoc water supply infrastructure planning processes by demonstrating the significant regret associated with the current baseline planned Friant-Kern Canal rehabilitation partnership, which yields substantially lower water gains and much higher financial risk for some partner districts compared to the Compromise Partnership. In practice, infrastructure partnerships are generally established via pre-existing relationships or ad hoc processes. Most infrastructure investments are then evaluated using low-resolution models that fail to capture key system features (i.e., interdependent flood-drought dynamics, institutional constraints, infrastructure operations, etc.), and the preferred alternative is selected based on highly aggregated traditional expected benefit-cost analyses. These processes fail to grapple with

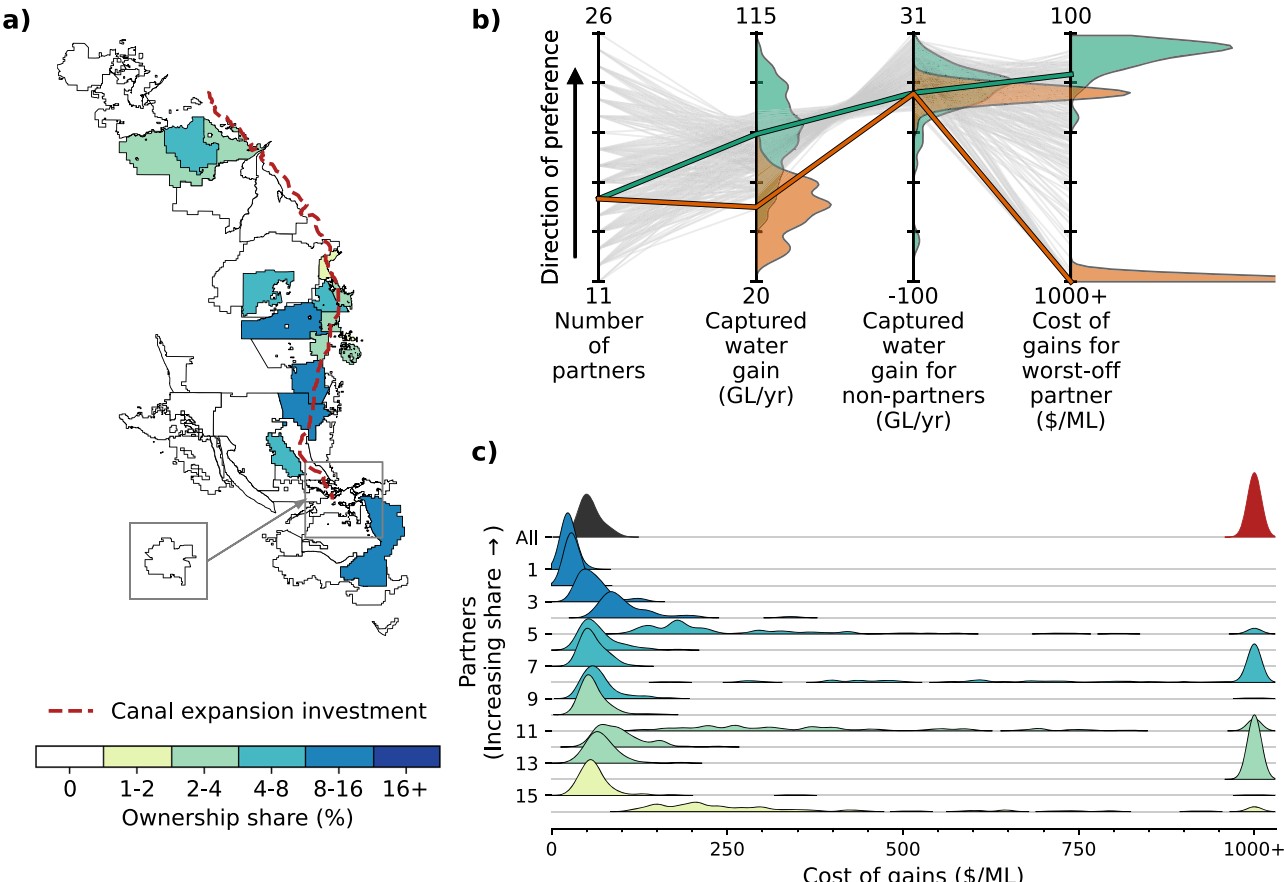

**Fig. 5 | Uncertain and heterogeneous performance of the Status Quo Partnership. a** Map showing the Friant Contractor water providers participating in the Status Quo Partnership, which invests only in canal expansion. Partners are colored by their ownership share in the partnership, while non-partners are colored white. **b** Parallel coordinate plot showing partnership performance across four conflicting objectives. Each gray line represents the performance of a different optimal tradeoff partnership, aggregated across the 64 sampled 30-year daily hydrologic sequences. The green line represents the aggregated performance of the Compromise Partnership, while the green shaded areas show the probability distribution of single-scenario performances for the Compromise Partnership. Similarly, the orange line and orange shaded areas represent the aggregated and disaggregated performance for the Status Quo Partnership. **c** Heterogeneity of partner-level performance for the Status Quo Partnership. Each shaded area labeled 1–16 represents the distribution of costs of gains across the 64 scenarios for a different water provider partner. The partners are sorted and colored by ownership share. The distributions labeled "All" represent the partnership-level aggregated costs (black) and the costs for the worst-off partner across alternative scenarios.

the complex tradeoffs, uncertainties, and heterogeneities in modern interconnected water supply systems and, therefore, are ill-equipped to design robust and equitable infrastructure partnerships. To elucidate the regret associated with current baseline planning processes, we analyze the ongoing rehabilitation of the Friant-Kern Canal by the Friant Contractors, a collection of water providers with contracts to receive Central Valley Project water from Millerton Lake via the Friant-Kern Canal (Fig. 1)[48–50]. We model a "Status Quo Partnership" (Fig. 5a) after this real-world example, as opposed to the partnerships in the optimal tradeoff set, which are designed via our multiobjective intelligent search process (see Methods).

Although it does provide significant benefits to its 16 partners, the Status Quo Partnership is matched or outperformed on all four aggregate performance metrics by the Compromise Partnership (Fig. 5b). The Status Quo Partnership produces 37% lower captured water gains on average, and when accounting for the uncertainty it is outperformed in the large majority of hydrologic scenarios. The Status Quo Partnership also produces slightly larger negative impacts on non-partners in the region on average (0.9 vs. 0.5 ML/year), although the Compromise partnership produces larger negative impacts in its most extreme hydrologic scenarios.

The Status Quo Partnership also carries significantly greater financial risk. The worst-partner cost of gains exceeds $1000/ML in all

79 sampled hydrologic scenarios. Two water providers in particular (Partners 8 and 14) receive marginal or even negative captured water gains, and thus experience costs over $1000/ML, in a large majority of scenarios (Fig. 5c, Supplementary Fig. S9). Five other providers also exceed $1000/ML in a smaller set of scenarios. In contrast, there are no sampled hydrologic scenarios where any of the 17 water providers in the Compromise Partnership experience costs of gains above $407/ML, and in 90% of scenarios, the worst-off partner pays less than $180/ML.

We also further test the robustness of the Compromise and Status Quo Partnerships using an ensemble of 20 downscaled CMIP5 climate models over the 2021–2050 period. Interestingly, both partnerships show relatively similar expected performance and levels of uncertainty for the CMIP5 ensemble compared to the original ensemble of synthetically generated scenarios (Supplementary Figs. S10 and S11). In fact, each partnership shows slightly improved captured water gains in the CMIP5 scenarios, as well as somewhat lower partner-level financial risks on the whole. However, the difference across ensembles for each partnership (Supplementary Figs. S10 and S11) is found to be much smaller than the differences between the two partnerships (Fig. 5), suggesting that the choice of ensemble in this case is unlikely to change the decision makers' preferences about which partnership is preferable.

Three major differences in partnership design contribute to the Compromise Partnership's dominance over the Status Quo Partnership. First, it couples the canal expansion project with new groundwater banking facilities, which help to capture more surplus water during high-flow periods when all partners' immediate demands are already satiated. A majority (83%) of partnerships in the optimal tradeoff set invest in both projects (Fig. 2a), suggesting that the performance gains from the synergistic pairing are generally worth the higher price tag. Second, although the Compromise Partnership and the Status Quo Partnership have significant overlap in participating providers (Figs. 4a and 5a), the Compromise Partnership discovered through multiobjective intelligent search removes several Friant Contractors with marginal benefits and significant financial risks. It also adds several other non-Friant providers that stand to benefit from the infrastructure investment. Widening the net beyond the Friant Contractors allows for a more diversified portfolio of water supplies and demands and increases the utilization of the infrastructure across a range of seasons and conditions. Lastly, the ownership shares and annual debt payments as refined through the multiobjective intelligent search process are better matched to partner-level captured water gains in the Compromise Partnership than the Status Quo Partnership, which helps to equalize the cost of gains across partners. These results highlight the significant regret associated with traditional ad hoc partnership design processes and the substantial improvements that can be achieved by combining detailed ensemble water supply modeling with multiobjective intelligent search.

## Discussion

According to the American Society of Civil Engineers, the U.S. will need to invest $109 billion per year in water infrastructure over the next 20 years to overcome the current investment gap in light of the deteriorating condition of the existing infrastructure stock as well as factors related to climate change, economic change, and regulatory change[5]. Collaborative partnerships and regionalization can be effective tools for facilitating investment by lowering costs and increasing flexibility. However, collaboration also brings new challenges related to the design and evaluation of robust infrastructure investments in institutionally complex water supply networks. Addressing these challenges will be critical to achieving California's water supply resilience and groundwater sustainability goals[21,42]. More broadly, these insights should be highly relevant to many other regions around the world that are working to adapt to increasing water scarcity, especially in regions with complex water institutions and interconnected infrastructure systems.

Our results have several key implications for water providers seeking to meet their supply reliability and groundwater sustainability goals through collaborative investment. First, water providers and planning agencies should assess financial risk as a key metric for infrastructure partnership design, along with more traditional planning metrics such as water supply reliability. Second, although the success of large partnerships (11–26 partners) in our analysis supports a collaborative vision for infrastructure planning, our results also highlight an important tension between partnership size and partner-level financial risk that must be navigated by water providers and planning agencies. Third, given the strong tradeoff between water supply benefits for partners vs. non-partners, there is a need for state and national planning agencies to provide more guidance on the role of external stakeholders in the planning process when infrastructure investments have the potential for negative external impacts. Fourth, the high level of variability across alternative sampled hydrologic scenarios demonstrates the importance of considering internal hydroclimatic variability in infrastructure partnership evaluation, while the significant partner-level heterogeneities highlight the critical role of highly detailed water supply models that can resolve complex infrastructural and institutional constraints and multiscale system dynamics. Both uncertainty and heterogeneity provide highly decision-relevant information that would not be available using traditional aggregated mean benefit-cost analyses. Fifth, given the preponderance of partnerships investing in both canal expansion and groundwater banking within the optimal tradeoff set, there is clearly a need for more integrated planning efforts that consider the interactive effects of multiple simultaneous infrastructure investments spanning local-to-regional scales within the larger water supply network. Finally, the severe performance tradeoffs that emerge from relatively subtle changes in partnership design across the optimal tradeoff set and the dominance of the Compromise Partnership over the Status Quo Partnership both demonstrate the value of multiobjective intelligent search for improving the design of infrastructure partnerships in institutionally complex water supply networks relative to traditional ad hoc planning processes.

In this study, we explore hydrologic uncertainty using a broad ensemble sampling of plausible 30-year daily streamflow sequences from a synthetic generator trained on historical observations. This represents an actively optimistic framing because we only account for stationary hydroclimatic variability while neglecting nonstationary climate change. Despite the optimistic framing, we nevertheless find that uncertainties in assessing water supply benefits and financial risks are highly consequential as a result of California's highly variable climate. We also find that our synthetically sampled streamflow scenarios span a similarly wide range of extremes as a multi-model downscaled CMIP5 ensemble for the 2021–2050 period, and that simulated partnership performance across the two ensembles is similar. Future work will consider the impact of a wider range of plausible nonstationary climate futures to account for the impact of climate change on uncertainty and decision-making. Additionally, it will consider other uncertainties related to future water demand, reservoir operations, infrastructure performance, and cost. These factors are expected to further widen the envelope of uncertainty around aggregate and partner-level impacts and exacerbate the challenge of designing robust partnerships.

Our results highlight the significant regret associated with traditional ad hoc partnership planning processes and the substantial improvements that can be achieved by combining detailed ensemble water supply modeling with multiobjective intelligent search. However, in practice, partnerships emerge as part of a complex human process within a broader historical and institutional context. Future work should investigate how these computational tools can best be integrated with more traditional stakeholder-based collaborative planning efforts in order to discover new alternatives, illuminate tradeoffs, and improve transparency to break down historical silos[80–83]. Local planners and operators have important knowledge about infrastructural and institutional behaviors, as well as legal, fiscal, and personal constraints, that may not be presently captured in our modeling framework. Thus, in practice it would be critical to complement the multiobjective intelligent search with iterative processes of evaluation and human feedback to better capture this explicit and implicit knowledge. It is also important to note that different water providers will have different abilities to pay for infrastructure, unique vulnerabilities to financial risk based on their underlying customer bases, and different levels of political power. Although we focus on equality between partners for this study, future work should extend this analysis by considering equity and asymmetries in economic and political power, customer vulnerability, racial injustice, and historical responsibility for groundwater overdraft[62,84–87]. The concepts of equity and resilience have many possible definitions, and more work is needed to understand how differing qualitative and quantitative understandings of these concepts are impacted by model representational fidelity and multiobjective formulations to shape perceptions of infrastructure performance and tradeoffs[88–91].

Future extensions of this work should also consider the role of contractual innovations, environmental impact bonds, engineering options analysis, and adaptive risk management strategies in redistributing and reducing the partner-level financial risks associated with cooperative infrastructure investments[34,77,92–94]. Lastly, we acknowledge that the complexity and computational requirements of our multiobjective intelligent search workflow may initially present a challenge for water providers looking to adapt this analysis for new contexts. This will also require breaking down silos across engineering, planning, and policy experts. University researchers, cooperative extension services, and federal/state planning agencies should invest resources towards developing significant capacities for facilitation, training, and technology transfer to enable the efficient application of cutting-edge computational tools to critical public planning challenges. The proliferation of free and open-source software and the declining costs of computing resources should also continue to reduce the barriers to this type of analysis over time.

## Methods

### Candidate infrastructure investments

Following Hamilton et al.[33], we consider two candidate infrastructure projects in this study. First is the rehabilitation of the Friant-Kern Canal. The canal has been severely damaged by subsidence related to groundwater overdraft, which has reduced its conveyance capacity by up to 60% in some regions[48,49]. A collection of water providers known as the Friant Contractors (Fig. 1) is currently investing roughly $50 million in rehabilitating the canal, with the remainder of the $500 million cost to be covered by various federal, state, and regional organizations[50]. For this study, we consider only the $50 million cost share allocated to the local water providers. We assume for simplicity that the modeled investment will restore the entire length of the canal to its original conveyance capacity.

The second candidate infrastructure project is a new groundwater banking facility near the confluence of the Friant-Kern Canal and the Tule River. This hypothetical project is not based on any particular existing or planned groundwater bank but is largely consistent with other recent and planned investments in the region[42]. We estimate that the bank's cost to project partners will be $50 million, which is on par with other recent and planned investments in groundwater recharge facilities[43]. There are three major parameters governing the capacity of groundwater recharge facilities in our water resource simulation model: the infiltration pond storage volume, the dry soil infiltration capacity, and the recovery pumping capacity. These are set to 0.74 GL/day (0.6 kAF/day), 0.37 GL (0.3 kAF), and 0.25 GL/day (0.2 kAF/day), ~25% of the parameter values for the largest groundwater bank in the region, the Kern Water Bank[95].

Both infrastructure projects are assumed to be financed using 30-year revenue bonds with 3% interest, broadly consistent with revenue bonds issued in 2019 by water districts in California[57]. Assuming equal annual payments over time, this translates to a debt payment of $2.55 million/year for partnerships investing in either canal expansion or groundwater banking, or $5.1 million/year for partnerships investing in both projects. All other infrastructure investment details are taken to be consistent with Hamilton et al.[33].

The elements of partnership design to be optimized can be understood as answering three questions. First, which infrastructure project should be built: canal expansion, groundwater banking, or both? Second, out of 40 candidate water providers in the region (Fig. 1), which subset should work together as partners? Third, once the set of partners has been established, how should the "ownership shares" be distributed amongst them? Each partner's ownership share dictates its share of priority capacity in the new infrastructure as well as its share of annual debt payment obligations, following Hamilton et al.[33]. Importantly, the annual debt payment cost considered here does not include any operating or maintenance costs associated with the infrastructure or the cost of procuring additional water. It thus represents an incomplete and optimistic view of the potential for financial risk associated with these infrastructure partnerships. The minimum allowable ownership share is set to 1%, so that each partner's share is between 1 and 100%. All of the partner ownership shares must sum to 100%.

### Infrastructure partnership evaluation

This study employs the California Food-Energy-Water System (CALFEWS), a free, open-source, Python/Cython-based simulation model of California's water resource system dynamics with a particular focus on the San Joaquin Valley[23]. The model combines a detailed representation of reservoir operations, water conveyance, interbasin transfer projects, water rights, municipal and agricultural demand, and conjunctive surface and groundwater supply management, including groundwater banking operations. CALFEWS operates on a daily time step and resolves water deliveries and operations at the level of individual water providers (e.g., municipal utilities, irrigation districts, and water storage districts). This spatiotemporal resolution combined with a detailed representation of water management institutions is unique among models for California and leads to improved performance in reproducing historical system behavior, from reservoir storages and releases to interbasin transfer project pumping to groundwater banking balances[23]. Most existing water supply planning models employ longer timesteps (e.g., daily or monthly), larger spatial aggregation (e.g., regional), and simplified representation of water management institutions and/or infrastructure constraints. CALFEWS' high-fidelity system representation allows for improved evaluation of infrastructure partnerships compared to lower-fidelity models that cannot resolve the multi-timescale dynamics of managed aquifer recharge or the multi-spatial scale distribution of water supply benefits and financial risks for local project partners (see Zeff et al.[23] and Hamilton et al.[33] for detailed discussions).

Hydroclimate in California is characterized by extreme interannual variability[51,52]. Over the last 20 years, the region has experienced three periods of persistent droughts that have been ended by extreme wet periods driven by atmospheric rivers. Facilitating robust management of the region's water supply system requires simulating operations under an array of scenarios that can represent current and plausible future hydroclimatic sequences in the region. To support this modeling effort, we generate synthetic full natural flows using a two-state Gaussian Hidden Markov Model (HMM), which has been shown to be an appropriate method to capture long-term persistent drought conditions[96,97]. The 110-year multi-site historical full natural flow reanalysis dataset described in Zeff et al.[23] is used for training and verification of the synthetic hydrologic scenario generator. The HMM consists of two "hidden" climate states representing wet and dry conditions. Within each of these states, log-space flows are sampled from respective Gaussian distributions. Following Thyer & Kuczera[98], we contribute a multi-site extension of the two-state HMM-based model to simultaneously generate full natural flows for 15 major watersheds in the Central Valley system. We generate 21 different 30-year sequences of multi-site correlated daily full natural flows to use within the multiobjective intelligent search step, along with another larger set of 79 additional 30-year daily streamflow sequences for the reevaluation step, so that each partnership in the final optimal tradeoff set has been evaluated on a total of 100 hydrologic scenarios of 30 years each (Supplementary Fig. S1). The number of sequences simulated in the search (21) and reevaluation (79) steps are selected to balance the fidelity of the noisy performance metrics against the significant computational expense of the search. The synthetic full natural flow records are well matched to the historical record in terms of correlation, distribution, and flow duration (Supplementary Figs. S6, S12, S13, S16). However, the synthetic ensemble also widens the envelope of extreme high and low flows beyond the historical record,

allowing for a deeper understanding of the impacts of internal hydroclimatic variability on infrastructure performance uncertainty. Finally, for two example partnerships, we also simulate performance under 20 downscaled CMIP5 climate models over the 2021–2050 period as an additional check on partnership robustness[53,99]. More details on the hydrologic scenario generator and the downscaled climate change scenarios can be found in Supplementary Note S1. Given a multi-site full natural flow record (e.g., from our synthetic scenario generator), CALFEWS uses an internal statistical reconstruction module to extrapolate inflows, snowpack levels, and other relevant hydroclimatic variables based on monthly linear regression relationships derived from historical observed data (see Zeff et al.[23] for a full description).

Each candidate infrastructure partnership is evaluated according to four decision-relevant metrics: (1) the number of water provider partners participating in the partnership; (2) the captured water gain, defined as the expected increase in total surface water deliveries across all project partners; (3) the captured water gain for non-partner water providers in the region; and (4) the cost of gains for the worst-off partner, defined as the annual debt payment divided by captured water gain for the worst-off partner in a partnership. The ideal partnership in our formulation would maximize Objective Metrics 1–3 and minimize Objective Metric 4, but given their conflicting nature, tradeoffs exist. These tradeoffs emerge because performance gains in any single objective come at the cost of giving up performance in one or more of the remaining objectives.

Each candidate infrastructure partnership is simulated and evaluated across 21 different 30-year daily hydrologic sequences from the synthetic generator in the multiobjective intelligent search step, and optimal tradeoff partnerships are then reevaluated across 79 additional 30-year sampled sequences to ensure robust generalization (see next section). Objective Metrics 2–3 are calculated by taking the expected value across sampled hydrologic scenarios and the sum across water provider partners (or non-partners in the case of Metric 3). Objective Metric 4 is calculated using the 90th percentile across sampled hydrologic scenarios and the max (i.e., worst-case) across partners in order to favor partnerships that can provide robust affordable water supply benefits across a wide range of plausible hydrologic conditions[89,91]. Note that we do not include the expected value of the cost of gains as an independent metric because it is monotonically related to the expected captured water gains metric for a given infrastructure investment, and therefore, it would not introduce significant independent information or tradeoffs to the decision formulation. The 90th percentile worst-partner formulation draws from previous efforts to define robust multiobjective problem formulations in noisy water resources simulation-optimization contexts[35,89,91]. More details on objective formulations can be found in Supplementary Note S2 and Supplementary Table S4.

## Multiobjective intelligent search

The Borg MOEA is a hyper-heuristic intelligent search algorithm employing auto-adaptive search operator selection, epsilon-dominance archiving, epsilon-progress stagnation detection, adaptive population sizing, and other beneficial features[58]. The Borg MOEA has been successfully employed across a range of difficult problems in water resources and engineering design characterized by high-dimensional, noisy, multimodal search spaces[100–105]. We employ the master-worker parallel version of the Borg MOEA, which is highly scalable on high-performance computing clusters[106,107].

The Borg MOEA is used to discover the set of optimal tradeoff partnerships (i.e., non-dominated or approximately Pareto-optimal solutions) within our multiobjective intelligent search procedure (Supplementary Fig. S1). We run four independent searches with different random seeds to account for the stochastic path-dependent nature of metaheuristic search algorithms[59]. The problem formulation that maps from the numeric decision vector that is optimized to the

underlying infrastructure partnership that is simulated is detailed in Supplementary Note S3. For each seed, we run the parallelized Borg MOEA for 96 h across 32 AMD EPYC 7742 nodes (4032 cores total) on the Pittsburgh Supercomputing Center's Bridges-2 supercomputer. In total, we test ~300,000 candidate infrastructure partnerships across the 4 seeds. The number of seeds and wall clock per seed is selected to balance the noise across seeds vs. the convergence, consistency, and diversity of solutions within each seed according to common multi-objective search metrics such as hypervolume, additive epsilon indicator, and generational distance[59,100,101]. Each seed exhibits acceptable convergence, consistency, and diversity given the constraints of our computing budget (Supplementary Fig. S17).

Upon completion, we calculate the final non-dominated reference set for each of the four randomly seeded optimization trials, yielding 1146 unique infrastructure partnerships. Each of these partnerships is then reevaluated across 79 additional 30-year synthetic hydrologic sequences outside of the training set to ensure that partnerships are robust in the face of a range of plausible hydroclimatic conditions. The final optimal tradeoff set is calculated as the set of partnerships from any of the four trials that are non-dominated based on reevaluated performance. This final Pareto set contains 270 unique infrastructure investment partnerships that represent the optimal tradeoffs across the four objectives. We also evaluate the Status Quo Partnership (see next section) with respect to the same 79 hydrologic scenarios for comparison. All performance metrics reported in this study refer to performance in the reevaluation step. The epsilon-dominance parameters that define the meaningful resolution for each objective with respect to the non-dominance calculation in the MOEA search and reevaluation steps can be found in Supplementary Table S4.

## Selection of Status Quo & compromise partnerships

We highlight two alternative partnerships in more detail in the results. The Status Quo Partnership (Fig. 5) is based on the real-world coalition of Friant Contractor water providers (Fig. 1) that are currently investing ~$50 million to rehabilitate the Friant-Kern Canal[49,50]. We design this partnership to mirror the existing institutional arrangement rather than being designed by our multiobjective intelligent search process. Given that the formal plans for apportioning cost shares across the partnership are not available, for the purpose of this analysis, the ownership share for each of the 16 partners is assumed to be proportional to its average historical delivery of Central Valley Project Friant water, following Hamilton et al.[33].

By comparison, the Compromise Partnership is selected as a representative solution from the broader 270-member optimal tradeoff set that outperforms the Status Quo Partnership and performs relatively well across all four objectives. This partnership is selected through a two-step process (Supplementary Fig. S18). First, we find the subset of optimal tradeoff partnerships that perform equal to or better than the Status Quo Partnership on all four aggregate objective metrics. The Compromise Partnership is then selected out of this subset by finding the partnership with the lowest worst-partner cost of gains.

## Data availability

Links to all supporting software and datasets for this paper, along with instructions for reproducing the analysis, can be found in the following metarepository on GitHub (https://github.com/IMMM-SFA/hamilton-etal_2024_naturecommunications). This metarepository is also permanently archived on Zenodo (https://doi.org/10.5281/zenodo.12801237)[108].

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

## Acknowledgements

Partial funding for this work was provided by the U.S. National Science Foundation (NSF), Innovations at the Nexus of Food-Energy-Water Systems, Track 2 (Award 1639268; G.C., P.R.), as well as the U.S. Department of Energy (DOE) Office of Science within the Earth and Environmental Systems Modeling Program. The DOE support is part of the Integrated Multisector, Multiscale Modeling Scientific Focus Area (Award DE-AC05-76RL01830; P.R., G.C.). The authors also acknowledge support from the Advanced Cyberinfrastructure Coordination Ecosystem Services and Support (ACCESS), which is supported by NSF grants ACI-2138259/2138286/2138307/2137603/2138296, and the Extreme Science and Engineering Discovery Environment (XSEDE), which is supported by NSF grant ACI-1548562. The multiobjective intelligent search and reevaluation simulation steps were carried out on Bridges-2 at the Pittsburgh Cybercomputing Center (PSC) through ACCESS allocation EES230083 (P.R.). Earlier preliminary optimization and simulation trials, which were foundational to this work, were carried out on Stampede2 at the Texas Advanced Computing Center (TACC) through XSEDE allocation TG-EAR090013 (P.R.). The authors also acknowledge the Cornell University

Center for Advanced Computing (CAC) for additional HPC resources and support. The views expressed in this work represent those of the authors and do not necessarily reflect the views or policies of the NSF, DOE, PSC, TACC, or CAC. We further acknowledge the World Climate Research Program's Working Group on Coupled Modeling, which is responsible for CMIP, and the climate modeling groups listed in the Supplementary Information of this paper for producing and making available their model output.

## Author contributions

A.H. conducted experiments, analyzed data, and wrote the first draft of the manuscript, except for figures and text contributed by R.G. related to the synthetic hydrologic scenario generator. A.H., R.G., and H.Z. contributed to code development. P.R. and G.C. supervised the project and contributed to the development of the paper's central ideas along with A.H. All authors participated in the final version of the manuscript.

## Competing interests

The authors declare no competing interests.
