## [Peer Review File · Nature Communications]

Resilient water infrastructure partnerships in institutionally complex systems face challenging supply and financial risk tradeoffsReviewers' Comments:

Reviewer #1:

Remarks to the Author:

Resilient water infrastructure partnerships in institutionally complex systems face challenging supply and financial risk tradeoffs

Main contributions

The paper uses evolutionary multi-objective optimisation and exploratory modelling methods to explore trade-offs and inform water infrastructure investment partnership design under various uncertainties related to climate, water demand, and institutions. The paper presents a case study in California, but the methodology can be applied globally to similar problems.

General feedback

Methods:

I have read the previous excellent technical papers from the author team and I am familiar with their methodology. I read this paper's methods. I have no doubts about its soundness and robustness, so I will not comment on it.

Visualisation:

The visualisations are stunning - simple and clear.

Structure:

The paper is well-written, but I have some suggestions for improvement:

First, the first subsection under Results (L101-161) does not seem to belong there. It looks more like a description of methods. I would either move it to Methods at the end or summarise it in a subsection under Introduction and before Results, if the journal format allows that.

Second, some of the other subsections under Results begin with a generic or a methodological sentence. For example, line 285 says "We also find that hydroclimatic variability leads to significant uncertainty in the performance of alternative infrastructure partnerships". This is kind of obvious and does not convey the main message of the subsection. I would personally start this and other subsections with the most important finding or insight that I have, so I can capture the attention of the readers right away. For instance, for the subsection starting at line 285, I would say something like "We found that uncertainty from various sources can have significant impacts on financial risks and debt consequences for water infrastructure, hence uncertainty should be systematically analysed in partnership design". The same comment applies to subsections starting at line 163 and line 369.

Discussion:

Partnerships, trade-offs, and compromises emerge within a complex institutional and political context with asymmetrical power dynamics among different actors. While this paper provides valuable insights for a robust design of such partnerships, it would be nice to also discuss some of the challenges and opportunities of implementing such a design in practice. For example, partnerships in practice may require the relevant institutions, jurisdictions, and various levels of

government to be better coordinated and aligned to support a robust and equitable outcome among various actors. Breaking down the silos between different policy areas and inter-sectoral programs can help minimise trade-offs and maximise synergies.

Reviewer #2:

Remarks to the Author:

The paper presents both a novel modeling framework that combines detailed water resource system modeling with multiobjective intelligent search and an empirical application of this modeling approach to infrastructure investment partnerships in California. The paper is divided in three main parts, the first part (i.e, introduction) motivates the need for this modeling approach and states the complexities and uncertainties that water planning communities face for managing water resources and designing infrastructure investment partnerships. Part two (i.e., results) presents results of the computational experiments used in the analysis, focusing on describing the multidimensional performance of different partnerships across the experimental ensemble. Part three (i.e, discussion) summarizes the main findings and discusses policy implications. The paper is very innovative and on a very interesting topic. Analyzing these partnerships considering both the financial and institutional specificity of the problem is very useful and does show a novel application of the methods put forward in the paper. However, after careful analysis of this paper, I find that there are various precisions, clarifications and potential improvements that would make this paper more impactful.

In the following lines, I make general recommendations and suggestions that summarize the specific comments and suggestions that I have made in the accompanying annotated pdf version of the paper. I believe these observations need to be addressed before publication.

My main concern relates to the hydroclimatic ensemble used in the study. The ensemble used in the study does not account for nonstationary long-term climate change impacts on temperature and precipitation. This creates an optimistic bias (e.g., not considering potentially very low and high temperature futures) in the climate ensemble used in the analysis. I think this is potentially problematic because of two reasons. The first that we have downscaled CMIP5 or CMIP6 climate scenarios available that are often used for climate vulnerability assessments. Thus, it is not clear why the authors decided not to use these ensembles? Is it because these are incompatible with the modeling framework? If that is the case, then wouldn't this make the modeling framework used in the analysis severely limited, regardless of the granularity of the model?. The second, and more relevant concern, is that I think the results of the analysis may change when considering adverse climate scenarios (e.g., severe declines in precipitation). For example, I think it is plausible to think that under adverse climate conditions, performance heterogeneity of partnerships may be reduced (e.g, since all agents face dire conditions, additional supply may impact the different performance metrics more homogeneously). If this is not the case, then, I think it would still be relevant to provide empirical evidence on this point since it would make the paper even more relevant as the performance heterogeneity found across partners would be invariant to climate change. In either case, it is probably necessary to introduce this long-term effects of climate change in the analysis or provide a stronger motivation to why these were included and elaborated on the potential implications that this methodological decision has on the results presented in the paper.

In the annotated version of the paper I have made various comments regarding the level of specificity of some of the evidence put forward in the analysis. In particular I think it would be important for readers to understand in more detail the classification thresholds used to classify the different behaviors discussed in the paper for example, what constitutes a “large partnerships” and what constitutes a “large ownership”. I think adding this level of clarification in several places in the paper can increase clarity of the paper. Additionally, I have made a few recommendations in the paper in sections in which I believe might be useful to run scenario discovery or a similar statistical learning technique to describe more explicitly the behavior observed in the simulations. For example, in its current form, it is not clear how climate, financial and institutional conditions interact to determine the performance of the partnerships across the different metrics. I believe this is valuable because there might be common partner characteristics in the groups that fair worse with the partnerships, and this might be valuable to report in the analysis. Overall, I think this is a great paper, well written and well documented, I hope the authors find my review useful to improve the quality of the paper.

Reviewer #3:

Remarks to the Author:

This study illustrated the importance of water infrastructure partnership design in terms of water gains to partners and non-partners, groundwater pumping, and financial risk. It is a well-written manuscript that points out an important yet under-researched consideration in water resources management.

There are a few questions that came up in my reading that I think would improve the manuscript if addressed.

First, I understand a main goal is to show the differences between the robust, multi-objective decision problem formulation vs a more aggregated but traditional benefit-cost analysis, but it still seems strange that an estimate of the central tendency of cost of water gains is not included as a decision metric in the MOEA. If this is because it is too similar to other metrics (e.g., captured water gain), that would be helpful for the reader to understand.

Second, it is unclear why the number of partners would need to be maximized, especially since you are already accounting for maximizing the gains for both partners and non-partners. The authors explain this a little at lines 42-45, but it seems unlikely that more partners would always be better across the board. Surely there is some cost to getting all the partners to agree on contracts and other requirements.

Third, some more background into the legal constraints behind water infrastructure partnerships in California would be helpful for the reader to understand what losses to non-partners might be acceptable. The authors say that "more guidance is needed from the state" but there must be some historical information on lawsuits to help understand what the threshold for legal action would be in practice.

Fourth, some of the results seem under-discussed. For example, what characteristics lead to some providers participating and/or carrying large shares of the project cost? Similarly, what characteristics lead to a partner paying more than their "fair share"?

Finally, the authors make a convincing argument for the need for more complex modeling and decision-making to improve water infrastructure partnership design, but it seems unlikely that water utilities would be able to do this on their own (run BORG, for example). And the complexity will be amplified when more than two projects are considered. Should utilities partner with universities to do this? Some more discussion on the feasibility of this argument would be beneficial.

Response to Reviewers for submission to Nature Communications: "Resilient water infrastructure partnerships in institutionally complex systems face challenging supply and financial risk tradeoffs"

We would like to thank the Editor and 3 anonymous Reviewers for their thoughtful comments, questions, and suggestions. Responding to them has improved the quality of our paper and we appreciate the time and effort put into this review.

In addition, we must note that during the process of drafting this response to reviews, we discovered a groundwater banking accounting error within our CALFEWS model. A brief description of the problem and confirmation of our fix is provided in Appendix A at the end of this document.

This error was not directly related to any of the reviewers' feedback, but after doing our due diligence we determined that it was consequential enough to require a full rerun of our computational experiment after fixing the issue. Due to the significant computational demands of the experiment in this paper, this was a rather long process and required attaining additional supercomputing allocations, so we appreciate the patience of the editor and reviewers for allowing us to take the time to handle this properly. Our correction however only impacted a few minor details of the results, and the overall figures and key takeaways from this new experiment are strongly consistent with the previous version. Consequently, we have been able to build off of the previous manuscript with clearly documented tracked changes, rather than drafting a new manuscript.

After fixing the error in our model chain, the main differences between the previous and current versions that we would like to bring to the reviewers' attention are as follows:

1. Building on our prior experience, we were able to refine the problem formulation and computational budget for this experiment. We decided to rerun the multiobjective intelligent search with 4 rather than 5 objectives. The groundwater pumping objective was removed because it aligned closely with the captured water gain objective where it is collinear, redundant, and not meaningfully conflicting. Thus, the groundwater pumping objective was not found to deliver significant additional insights for our original experiment and manuscript. This change in formulation has the advantage of reducing the overall number of function evaluations that needed to be run due to the reduced dimensionality of the Pareto-optimal solution set. Supplementary Fig. S17 demonstrates the strong convergence, diversity, and consistency of the search results. Thus, we have removed the groundwater pumping objective from the figures and discussion. As shown in our Tracked Changes revisions, this problem formulation change has not changed the key results and insights from the study.
2. In the original experiment, we explored two different decision variable formulations due to uncertainty in performance for the multiobjective search. That experiment found conclusively that the second formulation, which applies binary switch variables to include or exclude each partner, was the superior performing formulation that led to nearly all of the Pareto-optimal solutions in the reference set. In this revised follow-up experiment, we used this superior formulation exclusively to maximize the value of our computing allocation. We have removed discussion of the first formulation from the manuscript. This has the added advantage of increasing narrative clarity in the revised manuscript.
3. The number of random seed replicate search trials (4) and the number of hydrologic scenarios tested in the search phase (21) and the reevaluation stage (79) were updated to maximize informational value for our computing allocation based on the experience gained from the

original experiment. As noted above, Supplementary Fig. S17 demonstrates the strong convergence, diversity, and consistency of the results using these new parameters.

4. The results are largely consistent with the original manuscript. We again compare a “Compromise Partnership” to the “Status Quo Partnership”. The Compromise Partnership in this new draft is a different partnership than in the original draft since the underlying set of solutions has changed – but was selected using the same criteria and exhibits similar relative performance within the optimal tradeoff set.
5. One interesting difference in the results is that while the original optimal tradeoff set had 99% of partnerships investing in both canal expansion and groundwater banking and the remaining 1% investing only in groundwater banking, the new tradeoff set has 83% of partnerships investing in both projects and the remaining 17% investing only in canal expansion. This difference resulted from the error that we fixed in the CALFEWS model, which previously caused an overestimation of the benefits of groundwater banking under certain unusual circumstances which the intelligent search was able to exploit.
6. The other meaningful difference is that the minimum partnership size in the new optimal tradeoff set is 11 partners, compared to 3 partners in the original study. This result is also a result of fixing the error in the original model that overestimated the value of groundwater banking – in the original solution set, the smallest partnerships were generally those that invested only in groundwater banking. This is now reflected in how we discuss the impact of partnership size in the manuscript’s results and discussion sections.

In our response below, reviewer comments are shown in *blue italics*. This includes text excerpts included as part of reviewers’ comments. Line numbers and figure numbers in the reviewer comments (blue italics) refer to the original manuscript rather than the revised manuscript.

Our responses to reviewer comments are shown in regular black font. When including excerpts from the revised manuscript in our response, they will be indented and shown in **orange font**. All line numbers and figure numbers in our responses refer to the revised manuscript rather than the original manuscript. All text edits can be found in the manuscript submission file with tracked changes.

Reviewer #1 (Remarks to the Author):

Resilient water infrastructure partnerships in institutionally complex systems face challenging supply and financial risk tradeoffs

Main contributions

The paper uses evolutionary multi-objective optimisation and exploratory modelling methods to explore trade-offs and inform water infrastructure investment partnership design under various uncertainties related to climate, water demand, and institutions. The paper presents a case study in California, but the methodology can be applied globally to similar problems.

General feedback

Methods:

I have read the previous excellent technical papers from the author team and I am familiar with their methodology. I read this paper's methods. I have no doubts about its soundness and robustness, so I will not comment on it.

Thank you to Reviewer #1 for taking the time to review our work. We are glad you found it to be interesting and clear. Please see below for our responses to your comments.

Visualisation:

The visualisations are stunning - simple and clear.

Thank you. We spent a lot of time on figure design so glad to hear you find them effective.

Structure:

The paper is well-written, but I have some suggestions for improvement:

Reviewer comment 1.1: First, the first subsection under Results (L101-161) does not seem to belong there. It looks more like a description of methods. I would either move it to Methods at the end or summarise it in a subsection under Introduction and before Results, if the journal format allows that.

Nature Communications does not allow subsections within the Introduction, and generally the Introduction focuses on high level study motivations and contributions. We decided to use the first subsection of the Results to give a basic overview of the study location and the methods. This section is much less detailed than the deeper treatment given in the Methods and the Supplementary Information, and is designed to give enough information for the reader to understand the context and importance of the Results. This structure is common in Nature Communications papers (see for example (Basheer et al., 2022; Cong et al., 2022; Fletcher et al., 2019; Gazzotti et al., 2021)). However, we are open to moving this material to the Introduction at the Editor's request.

Reviewer comment 1.2: Second, some of the other subsections under Results begin with a generic or a methodological sentence. For example, line 285 says "We also find that hydroclimatic variability leads to significant uncertainty in the performance of alternative infrastructure partnerships". This is kind of obvious and does not convey the main message of the subsection. I would personally start this and other subsections with the most important finding or insight that I have, so I can capture the attention of the readers right away. For instance, for the subsection starting at line 285, I would say something like "We found that uncertainty from various sources can have significant impacts on financial risks and debt consequences for water infrastructure, hence uncertainty should be systematically analysed in partnership design". The same comment applies to subsections starting at line 163 and line 369.

Thank you for this suggestion. We have revised the opening sentences of three results sections to highlight key messages as suggested:

Line 170: Our multiobjective intelligent search process yields a set of 270 unique partnership structures which are highly diverse in terms of their design (i.e., different combinations of partner districts and their levels of relative investment) as well as their performance across

different objectives – this broad suite of alternatives collectively represents a wealth of information which would not have been available using traditional ad hoc partnership planning processes which typically consider only a small number of candidate arrangements.

Line 332: We also find that uncertainty from hydroclimatic variability can have significant impacts on water supply and financial risk consequences for infrastructure investments, and thus uncertainty should be systematically analyzed in partnership design.

Line 423: Lastly, we highlight the potential to meaningfully improve existing ad hoc water supply infrastructure planning processes by demonstrating the significant regret associated with the current baseline planned Friant-Kern Canal rehabilitation partnership, which yields substantially lower water gains and much higher financial risk for some partner districts compared to the Compromise Partnership.

Discussion:

Reviewer comment 1.3: Partnerships, trade-offs, and compromises emerge within a complex institutional and political context with asymmetrical power dynamics among different actors. While this paper provides valuable insights for a robust design of such partnerships, it would be nice to also discuss some of the challenges and opportunities of implementing such a design in practice. For example, partnerships in practice may require the relevant institutions, jurisdictions, and various levels of government to be better coordinated and aligned to support a robust and equitable outcome among various actors. Breaking down the silos between different policy areas and inter-sectoral programs can help minimise trade-offs and maximise synergies.

Thank you for this comment. We have expanded our discussion of these issues at the paper's conclusion:

Line 545: Our results highlight the significant regret associated with traditional ad hoc partnership planning processes and the substantial improvements that can be achieved by combining detailed ensemble water supply modeling with multiobjective intelligent search. However, in practice, partnerships emerge as part of a complex human process within a broader historical and institutional context. Future work should investigate how these computational tools can best be integrated with more traditional stakeholder-based collaborative planning efforts in order to discover new alternatives, illuminate tradeoffs, and improve transparency to break down historical silos (Basdekas & Hayslett, 2021; Moallemi et al., 2021; Smith et al., 2019, 2022). Local planners and operators have important knowledge about the infrastructural and institutional behaviors, as well as legal, fiscal, and personal constraints, that may not be presently captured in our modeling framework. Thus, in practice it would be critical to complement the multiobjective intelligent search with iterative processes of evaluation and human feedback to better capture this explicit and implicit knowledge. It is also important to note that different water providers will have different abilities to pay for infrastructure, unique vulnerabilities to financial risk based on their underlying customer bases, and different levels of political power. Although we focus on equality between partners for this study, future work should extend this analysis by considering equity and asymmetries in economic and political power, customer vulnerability, racial injustice, and historical responsibility for groundwater

overdraft (Avelino, 2021; Dobbin & Lubell, 2021; Fernandez-Bou et al., 2021; Fletcher et al., 2022; Osman & Faust, 2021). The concepts of equity and resilience have many possible definitions, and more work is needed to understand how differing qualitative and quantitative understandings of these concepts are impacted by model representational fidelity and multiobjective formulations to shape perceptions of infrastructure performance and tradeoffs (Ciullo et al., 2020; Jafino et al., 2021; McPhail et al., 2018; Quinn et al., 2017).

Line 568: Lastly, we acknowledge that the complexity and computational requirements of our multiobjective intelligent search workflow may initially present a challenge for water providers looking to adapt this analysis for new contexts. This will also require breaking down silos across engineering, planning, and policy experts. University researchers, cooperative extension services, and federal/state planning agencies should invest resources towards developing significant capacities for facilitation, training, and technology transfer to enable efficient application of cutting-edge computational tools to critical public planning challenges. The proliferation of free and open-source software and the declining costs of computing resources should also continue to reduce the barriers to this type of analysis over time.

Reviewer #2 (Remarks to the Author):

The paper presents both a novel modeling framework that combines detailed water resource system modeling with multiobjective intelligent search and an empirical application of this modeling approach to infrastructure investment partnerships in California. The paper is divided in three main parts, the first part (i.e, introduction) motivates the need for this modeling approach and states the complexities and uncertainties that water planning communities face for managing water resources and designing infrastructure investment partnerships. Part two (i.e., results) presents results of the computational experiments used in the analysis, focusing on describing the multidimensional performance of different partnerships across the experimental ensemble. Part three (i.e, discussion) summarizes the main findings and discusses policy implications. The paper is very innovative and on a very interesting topic. Analyzing these partnerships considering both the financial and institutional specificity of the problem is very useful and does show a novel application of the methods put forward in the paper. However, after careful analysis of this paper, I find that there are various precisions, clarifications and potential improvements that would make this paper more impactful.

In the following lines, I make general recommendations and suggestions that summarize the specific comments and suggestions that I have made in the accompanying annotated pdf version of the paper. I believe these observations need to be addressed before publication.

Thank you to Reviewer #2 for taking the time to review our paper. We are glad you found it interesting and novel and appreciate the detailed comments and suggestions. Please see below for our responses.

Reviewer comment 2.1: My main concern relates to the hydroclimatic ensemble used in the study. The

ensemble used in the study does not account for nonstationary long-term climate change impacts on temperature and precipitation. This creates an optimistic bias (e.g., not considering potentially very low and high temperature futures) in the climate ensemble used in the analysis. I think this is potentially problematic because of two reasons. The first is that we have downscaled CMIP5 or CMIP6 climate scenarios available that are often used for climate vulnerability assessments. Thus, it is not clear why the authors decided not to use these ensembles? Is it because these are incompatible with the modeling framework? If that is the case, then wouldn't this make the modeling framework used in the analysis severely limited, regardless of the granularity of the model?. The second, and more relevant concern, is that I think the results of the analysis may change when considering adverse climate scenarios (e.g., severe declines in precipitation). For example, I think it is plausible to think that under adverse climate conditions, performance heterogeneity of partnerships may be reduced (e.g., since all agents face dire conditions, additional supply may impact the different performance metrics more homogeneously). If this is not the case, then, I think it would still be relevant to provide empirical evidence on this point since it would make the paper even more relevant as the performance heterogeneity found across partners would be invariant to climate change. In either case, it is probably necessary to introduce this long-term effects of climate change in the analysis or provide a stronger motivation to why these were included and elaborated on the potential implications that this methodological decision has on the results presented in the paper.

Thank you to Reviewer #2 for this important comment. The impact of climate change on our results is a concern that many readers will likely share, so we appreciate the opportunity to address it more fully. We will respond to the set of questions above in three related parts. First, we present new figures that compare the 100 synthetically generated 30-year streamflow scenarios from our experiment to the streamflow ensemble generated from 20 alternative downscaled CMIP5 models over the 1950-2016 period (consistent with the data used to train the synthetic generator) and the 2021-2050 period (consistent with the 30-year horizon used for investment debt in this work). We use these results to provide additional support for our original experimental design based on our focus on hydroclimatic internal variability as a key aspect of robust infrastructure design. Second, we present new results from rerunning the Compromise and Status Quo Partnerships under the 2021-2050 CMIP5 scenarios, and compare the expected performance and uncertainty ranges to those generated using our synthetic scenarios. Finally, we summarize the advantages of our approach and address the remaining questions enumerated above.

First, Supplementary Figs. 6-7 provide a comparison of the historical streamflow, synthetic ensemble, and downscaled CMIP5 ensemble.

Supplementary Fig. S6: Comparison of full natural flow (FNF) distributions over the historical period. Distributions are shown for (a) aggregated major surface water reservoirs north of Millerton Lake (Shasta, Oroville, New Bullards Bar, Folsom, New Melones, Don Pedro, McClure), (b) Millerton Lake, (c) and aggregated major surface water reservoirs south of Millerton Lake (Pine Flat, Kaweah, Success, Isabella). Each subfigure shows the non-exceedance curves for annual FNF. “Historical” refers to the 110-year historical full natural flow reanalysis dataset. “Synthetic” refers to the 100 different 30-year synthetic flow scenarios used during the multiobjective intelligent search and reevaluation steps. “Projections 4.5” and “Projections 8.5” refer to the 1950-2015 results from 10 downscaled CMIP5 models using RCP 4.5 and RCP 8.5.

Supplementary Fig. S7: Comparison of full natural flow (FNF) distributions over the 2021-2050 period. Distributions are shown for (a) aggregated major surface water reservoirs north of Millerton Lake (Shasta, Oroville, New Bullards Bar, Folsom, New Melones, Don Pedro, McClure), (b) Millerton Lake, (c) and aggregated major surface water reservoirs south of Millerton Lake (Pine Flat, Kaweah, Success, Isabella). Each subfigure shows the non-exceedance curves for annual FNF. “Historical” refers to the 110-year historical full natural flow reanalysis dataset. “Synthetic” refers to the 100 different 30-year synthetic flow scenarios used during the multiobjective intelligent search and reevaluation steps. “Projections 4.5” and “Projections 8.5” refer to the 2021-2050 results from 10 downscaled CMIP5 models using RCP 4.5 and RCP 8.5.

We have added the following text in the methods:

Line 655: Finally, for two example partnerships, we also simulate performance under 20 downscaled CMIP5 climate models over the 2021-2050 period as an additional check on partnership robustness (L. D. Brekke et al., 2009; Cohen et al., 2020). More details on the hydrologic scenario generator and the downscaled climate change scenarios can be found in Supplementary Note S1.

More detail is provided in the Supporting Information explaining the comparison of the synthetically generated scenarios to the CMIP5 scenarios:

Line 225: Supplementary Fig. S6 shows that the 100 synthetically sampled 30-year streamflow scenarios used in this study follow a similar annual non-exceedance probability curve compared to the 1906-2015 historical record which was used to train the synthetic generator. At the same time, the ensemble of synthetic scenarios is found to substantially widen the envelope of extreme high- and low-flows compared to the historical record. This is critical for accurately characterizing risk in long-lived infrastructure investments due to the critical impact of hydroclimatic internal variability in regions like California. The first 21 of these flow sequences are used in the optimization stage of the modeling framework and the other 79 flow sequences are used in the reevaluation phase.

In order to compare our synthetically generated streamflow ensemble to the streamflows that could be experienced under anthropogenic climate change over a 30-year investment horizon, we also consider an ensemble of downscaled climate change scenarios from the Climate Model Intercomparison Project Phase 5 (CMIP5) (Taylor et al., 2012). Brekke et al. (2013) previously released a suite of hydrologic scenarios for California generated by using downscaled CMIP5 simulations to drive the Variable Infiltration Capacity (VIC) hydrologic model (L. Brekke et al., 2013; Liang et al., 1994). Cohen et al. (2020) then aggregated and organized these results to get basin streamflows to drive a reservoir operations model for California (Cohen et al., 2020). We utilize a subset of streamflow scenarios from Cohen et al. (2020) for 10 CMIP5 models: CCSM4 (Gent et al., 2011), CNRM-CM5.1 (Volodire et al., 2013), CSIRO-Mk3.6.0 (Collier et al., 2011), GFDL-CM3 (Griffies et al., 2011), GFDL-ESM2M (Dunne et al., 2012), HADGEM2-CC (Martin et al., 2011), HADGEM2-ES (Martin et al., 2011), INMCM4 (Volodin et al., 2010), IPSL-CM5A-MR

(Dufresne et al., 2013), and MIROC5 (Watanabe et al., 2010). For each model, we use both RCP 4.5 and 8.5, for a total of 20 combinations.

Compared to the synthetically generated streamflow scenarios, the 20 downscaled CMIP5 scenarios show more limited range of variability and extremes over the 1950-2015 period (Supplementary Fig. S6). They also appear to show a consistent low bias in the driest years over the historical period. Over the 2021-2050 period, the downscaled CMIP5 scenarios are found to display a wider envelope of extreme annual behavior compared to the historical period (Supplementary Fig. S7). The envelope of extremes in this case is more similar to the synthetically generated scenarios. This shows that although the synthetic generator is trained on the historical record, it nonetheless is able to generate scenarios with similarly extreme behavior as found in the CMIP5 scenarios which explicitly account for anthropogenic climate change.

These results support the use of these synthetically generated scenarios given our focus on internal variability and designing for robust performance. More generally, recent research has shown that most general circulation models show poor representation of internal variability and uncertainty at regional scales, especially for precipitation (Jain et al., 2023; Lehner et al., 2020; Lehner & Deser, 2023). This uncertainty is critical to risk assessment and the design of robust infrastructure systems in highly uncertain conditions, which is the primary justification for the synthetic generation strategy employed in this work.

Next, we simulated both the Compromise and Status Quo Partnership across the 20 CMIP5 scenarios in order to test whether the synthetic ensemble and the climate change ensemble would lead decision makers to prefer different partnerships. Supplementary Figs. S10-11 show the results of this experiment.

Supplementary Fig. S10: Uncertain performance of Compromise Partnership in synthetic scenarios vs. CMIP5 scenarios. (a) Parallel coordinate plot showing partnership performance across four conflicting objectives. Each gray line represents the performance of a different optimal tradeoff partnership, aggregated across the 79 synthetic 30-year daily hydrologic sequences. The green line represents the aggregated performance of the Compromise Partnership across the synthetic scenarios, while the green shaded areas show the probability distribution of single-synthetic-scenario performances for the Compromise Partnership. For comparison, the blue shaded area represents the probability distribution of single-CMIP5-scenario performances for the Compromise Partnership. **(b)** Heterogeneity of partner-level performance for the Compromise Partnership across the synthetic scenarios. Each shaded area labeled 1-16 represents the distribution of costs of gains for a different water provider partner across the 79 sampled 30-year daily hydrologic sequences. The partners are sorted and colored

by ownership share. The distributions labeled "All" represent the partnership-level aggregated costs (black) and the costs for the worst-off partner across alternative scenarios. (c) Same as (b), but for the CMIP5 scenarios.

Supplementary Fig. S11: Uncertain performance of Status Quo Partnership in synthetic scenarios vs. CMIP5 scenarios. (a) Parallel coordinate plot showing partnership performance across four conflicting objectives. Each gray line represents the performance of a different optimal tradeoff partnership, aggregated across the 79 synthetic 30-year daily hydrologic sequences. The orange line represents the aggregated performance of the Status Quo Partnership across the synthetic scenarios, while the orange shaded areas show the probability distribution of single-synthetic-scenario performances for the Status Quo Partnership. For comparison, the blue shaded area represents the probability distribution of single-CMIP5-

scenario performances for the Status Quo Partnership. (b) Heterogeneity of partner-level performance for the Status Quo Partnership across the synthetic scenarios. Each shaded area labeled 1-16 represents the distribution of costs of gains for a different water provider partner across the 79 sampled 30-year daily hydrologic sequences. The partners are sorted and colored by ownership share. The distributions labeled "All" represent the partnership-level aggregated costs (black) and the costs for the worst-off partner across alternative scenarios. (c) Same as (b), but for the CMIP5 scenarios.

We added the following paragraph to the main text explaining these results.

Line 468: We also further test the robustness of the Compromise and Status Quo Partnerships using an ensemble of 20 downscaled CMIP5 climate models over the 2021-2050 period. Interestingly, both partnerships show relatively similar expected performance and levels of uncertainty for the CMIP5 ensemble compared to the original ensemble of synthetically generated scenarios (Supplementary Fig. S10-S11). In fact, each partnership shows slightly improved captured water gains in the CMIP5 scenarios, as well as somewhat lower partner-level financial risks on the whole. However, the difference across ensembles for each partnership (Supplementary Figs. S10-S11) is found to be much smaller than the differences between the two partnerships (Fig. 5), suggesting that the choice of ensemble in this case is unlikely to change decision-makers' preferences about which partnership is preferable.

We also expand the relevant section of the Discussion as follows:

Line 531: In this study, we explore hydrologic uncertainty using a broad ensemble sampling of plausible 30-year daily streamflow sequences from a synthetic generator trained on historical observations. This represents an actively optimistic framing because we only account for stationary hydroclimatic variability while neglecting nonstationary climate change. Despite the optimistic framing, we nevertheless find that uncertainties in assessing water supply benefits and financial risks are highly consequential as a result of California's highly variable climate. We also find that our synthetically sampled streamflow scenarios span a similarly wide range of extremes as a multi-model downscaled CMIP5 ensemble for the 2021-2050 period, and that simulated partnership performance across the two ensembles is similar. Future work will consider the impact of a wider range of plausible nonstationary climate futures to account for the impact of climate change on uncertainty and decision-making. Additionally, it will consider other uncertainties related to future water demand, reservoir operations, infrastructure performance, and cost. These factors are expected to further widen the envelope of uncertainty around aggregate and partner-level impacts and exacerbate the challenge of designing robust partnerships.

In summary, we chose to develop a synthetic streamflow generation approach based on a multi-site Hidden Markov Model in order to ensure a robust characterization of internal hydroclimatic variability. When comparing these sampled scenarios to a 20-member downscaled CMIP5 ensemble spanning RCP 4.5 and 8.5, we found that our synthetically sampled scenarios were able to cover a similar range of hydroclimatic extremes compared to the CMIP5 models over the 30-year horizon for this infrastructure

investment. Additionally, we find relatively minor differences between the synthetically sampled scenarios and the CMIP5 scenarios in terms of the three uncertain objective metrics for the Compromise and Status Quo Partnerships. The difference in ensemble thus does not represent a decision-relevant difference with respect to the preference for the Compromise Partnership over the Status Quo Partnership within the context of our decision formulation.

Regarding Reviewer 2's hypothesis that climate change could cause the partner-level impacts to be less heterogeneous: we do not find evidence supporting this. The results are quite similar in the CMIP5 scenarios vs. the synthetically sampled scenarios. Moreover, this hypothesis neglects the institutional side of the water rights landscape – partners will not necessarily experience the negative impacts of climate change equally. In fact, given the existence of prior appropriations and other hierarchical/asymmetric features of California's water institutions, it is likely that adverse climate change would lead to a "rich get richer" dynamic that amplifies risk heterogeneities rather than dampening them. However, the results of the present study cannot confirm not deny these hypotheses because we find marginal changes resulting from using the CMIP5 scenarios rather than the stationary ensemble.

Nevertheless, we do believe that further study is warranted to better understand the impact of these hydroclimatic non-stationarities on infrastructure partnership performance and financial risk. In ongoing work, we are further characterizing partnership robustness across a wider range of hydroclimatic uncertainties, as well as other uncertainties such as economic conditions and environmental regulations, as reflected in the last sentence of the paragraph quoted above (Line 531).

Reviewer comment 2.2: In the annotated version of the paper I have made various comments regarding the level of specificity of some of the evidence put forward in the analysis. In particular I think it would be important for readers to understand in more detail the classification thresholds used to classify the different behaviors discussed in the paper for example, what constitutes a "large partnerships" and what constitutes a "large ownership". I think adding this level of clarification in several places in the paper can increase clarity of the paper. Additionally, I have made a few recommendations in the paper in sections in which I believe might be useful to run scenario discovery or a similar statistical learning technique to describe more explicitly the behavior observed in the simulations. For example, in its current form, it is not clear how climate, financial and institutional conditions interact to determine the performance of the partnerships across the different metrics. I believe this is valuable because there might be common partner characteristics in the groups that fair worse with the partnerships, and this might be valuable to report in the analysis.

Thank you for this feedback. Please see below for our detailed responses to these individual questions and comments, as posed separately in more detail in the annotated PDF provided by Reviewer #2.

Overall, I think this is a great paper, well written and well documented, I hope the authors find my review useful to improve the quality of the paper.

Reviewer #2 (comments from annotated PDF)

For each comment in the annotated PDF, we have first copied the relevant text snippet from the original manuscript, followed by the reviewer's annotated comment related to that snippet, followed by our response to the comment.

Line 42: "Multiple water providers collaborating to finance, build, and operate shared facilities with joint benefits can be more cost effective than independent provisioning due to economies of scale, reduced redundancy, improved access to capital, and diversification of supply and demand patterns^{15–20}."

Reviewer comment 2.3: Less redundancies can lead to higher vulnerabilities

Thanks, this is a good point, but does not negate the point we are making. The statement made in the quote above is that reduced redundancy can lower costs, which is one reason for the popularity of joint partnerships. But we agree that this redundancy can lead to reliability tradeoffs, and have added the following line to the next paragraph on the risks of collaborative investments:

Line 59: Shared water supply facilities that reduce system redundancy can also increase vulnerability during extreme conditions.

Line 62: "Water supply planning models are often highly aggregated, with many water providers grouped together, which limits their ability to discover local-scale tradeoffs for individual partners^{23,33}"

Reviewer comment 2.4: Is there evidence on the impact that level of aggregation has on key performance indicators? Some models can be highly aggregated in terms of operation and climate, but maybe highly disaggregated in scale and agents? Can you elaborate more on why this is a limitation for your analysis? Does this mean that without a large-scale your analysis will be pointless?

Thank you for this comment. There are indeed many different dimensions of aggregation in a complex human-natural systems model, as discussed by Yoon et al., 2022. We make the point that the CALFEWS model used in this study represents a significant improvement along three dimensions of aggregation when compared to most existing water supply simulation models. First, many existing water supply simulation models use highly aggregated nodal structure that consolidate many local actors into a single pseudo node (i.e., combining different water districts that share infrastructure). This aggregation limits their ability to accurately characterize local-scale incentives, infrastructural/institutional constraints, and strategic behavior, and the emergent impacts of these local scale differences on broader system dynamics. A major advantage of the CALFEWS model used in this work is that it represents individual water districts as unique nodes characterized by different water rights, cropping patterns, and groundwater banking behaviors. A discussion of these issues and a detailed verification of CALFEWS' skill at representing regional and local behavior can be found in (Zeff et al., 2021).

Second, water supply models that aggregate many local districts into regional pseudo-nodes have limited ability to quantify performance and risks for individual local districts, thus providing limited value for decision makers at local districts which are trying to decide whether and how to participate in these infrastructure partnerships. CALFEWS' district-level scale allows us to provide detailed disaggregated

performance metrics at the scale needed for decision makers to evaluate these important water supply resilience partnerships (Hamilton et al., 2022).

Third is the related issue of temporal aggregation – many existing water supply planning models use a large timestep such as one month. This limits their ability to accurately characterize faster timescale dynamics such as peak flows during atmospheric river events, which is critical for accurately assessing the amount of available water that could be captured given engineered and institutional constraints for a planned infrastructure investment (Hanak et al., 2018; Kocis & Dahlke, 2017). A major advantage of the CALFEWS model used in this work is its daily timescale and its explicit focus on the coupled dynamics between fast-timescale peak flow events and long-timescale drought planning and groundwater banking dynamics (Zeff et al., 2021).

These topics are discussed in various places throughout the manuscript, including:

Line 615: This study employs the California Food-Energy-Water System (CALFEWS), a free, open-source, Python/Cython-based simulation model of California’s water resource system dynamics with a particular focus on the San Joaquin Valley (Zeff et al., 2021). The model combines detailed representation of reservoir operations, water conveyance, interbasin transfer projects, water rights, municipal and agricultural demand, and conjunctive surface and groundwater supply management including groundwater banking operations. CALFEWS operates on a daily time step and resolves water deliveries and operations at the level of individual water providers (e.g., municipal utilities, irrigation districts, and water storage districts). This spatiotemporal resolution combined with detailed representation of water management institutions is unique among models for California and leads to improved performance in reproducing historical system behavior, from reservoir storages and releases to interbasin transfer project pumping to groundwater banking balances (Zeff et al., 2021). Most existing water supply planning models employ longer timesteps (e.g., daily or monthly), larger spatial aggregation (e.g., regional), and simplified representation of water management institutions and/or infrastructure constraints. CALFEWS’ high-fidelity system representation allows for improved evaluation of infrastructure partnerships compared to lower fidelity models that cannot resolve the multi-timescale dynamics of managed aquifer recharge or the multi-spatial scale distribution of water supply benefits and financial risks for local project partners (see Zeff et al. (2021) and Hamilton et al. (2022) for detailed discussions).

Line 64: *“Moreover, little guidance exists to facilitate the design of resilient and equitable water supply infrastructure partnerships in institutionally complex systems.”*

Reviewer comment 2.5: How do you operationalize resilience or equity?

A major focus of our work is highlighting the many ways in which a more detailed and disaggregated simulation model combined with multiobjective intelligent search can open up new possibilities in the planning of resilient and equitable water infrastructure. As discussed by (Fletcher et al., 2022; Jafino et al., 2021), and related to our response to Reviewer comment 2.4 above, many water resources models and other human-natural systems models are highly aggregated which does not allow for local-scale

equity to even be assessed. Similarly, many studies do not adequately address uncertainty, which limits their ability to quantify resilience. A more resolved system representation is required to even allow for the possibility of quantifying or optimizing for equity and resilience.

In this paper, we use the terms resilient and equitable in rather broad terms so that our statements should remain valid beyond any particular precise definition. However, there are many different ways to quantify equity or resilience, which can lead to different results (Anderies et al., 2013; Avelino, 2021; Ciullo et al., 2020; Fletcher et al., 2022; McPhail et al., 2018; Osman & Faust, 2021). Additionally, there are other aspects to equity beyond model fidelity, such as representational or procedural equity. These are important considerations but are beyond the scope of the present work. However, we have expanded the future work section related to these topics:

Lines 555: It is also important to note that different water providers will have different abilities to pay for infrastructure, unique vulnerabilities to financial risk based on their underlying customer bases, and different levels of political power. Although we focus on equality between partners for this study, future work should extend this analysis by considering equity and asymmetries in economic and political power, customer vulnerability, racial injustice, and historical responsibility for groundwater overdraft (Avelino, 2021; Dobbin & Lubell, 2021; Fernandez-Bou et al., 2021; Fletcher et al., 2022; Osman & Faust, 2021). The concepts of equity and resilience have many possible definitions, and more work is needed to understand how differing qualitative and quantitative understandings of these concepts are impacted by model representational fidelity and multiobjective formulations to shape perceptions of infrastructure performance and tradeoffs (Ciullo et al., 2020; Jafino et al., 2021; McPhail et al., 2018; Quinn et al., 2017).

Line 66: *“Existing economic planning frameworks for partnership design and cost apportionment are generally based on expected value benefit-cost analysis and/or game theoretic methods^{36–40}These methods are not well suited to realistic systems characterized by a large number of potential partners, interactive network effects, high uncertainty, and multiple conflicting objectives.”*

Reviewer comment 2.6: Not sure I agree with this comment, why are these methods not well suited? all models are abstractions of reality, please explain the specific characteristics that would make this model not useful for real decision making? is this the level of detail? I think as long as the model is calibrated rigorously, this can serve as a modeling tool, is not that the case? for example, did you calibrate the model used for this study? if so, how that calibration is done such that the model represents realistically the decision space you are studying?

There are two parts to this question. First, CALFEWS moves well beyond an aggregate calibrated model by intentionally capturing in detail critical institutional, operational, and infrastructure systems dynamics in its structure. It is an adaptive operations, rules-based simulation engine built through careful representation of California’s water supply regulatory documents and engineering technical reports. CALFEWS has a high level of detail in its representation of engineering constraints, water rights and interbasin transfer contracts, and groundwater banking arrangements. The model has been tested extensively and found to capture historical system dynamics with a high degree of fidelity, including

observed reservoir levels, aqueduct pumping, water project deliveries, and groundwater bank balances – see (Zeff et al., 2021) for details.

Second, the text snippet referred to in this comment is not specifically a critique of water resource simulation models, but rather economic decision analytic frameworks commonly used to help plan and finance infrastructure partnerships (Bird & Slack, 2017; De Souza et al., 2011; Giglio & Wrightington, 1972; Madani, 2010; Watkins, 1998). Expected value benefit-cost analysis, which is commonly used to decide which project alternatives should be built, generally calculate benefits and cost on a highly aggregated level rather than considering partner-level impacts. They also typically do not consider uncertainty in a rigorous way. For example, the U.S. Bureau of Reclamation Feasibility Report (U.S. Bureau of Reclamation, 2020) for the ongoing Friant-Kern Canal Rehabilitation Project used a highly simplified representation of total regional water supply benefits that would result from different project alternatives, without consideration of district-level behavior, impacts, and tradeoffs. They also did not consider a full range of long-term hydroclimatic conditions, instead reporting expected value benefits and performance in a small number of alternative water year types.

Game theoretic methods for cost apportionment among partners are typically based on combinatorial search over all combinations of partners, and cannot scale beyond a small number of partners due to the factorial explosion of combinations – this is a non-starter for a large-scale partnerships like the present work, which has 40 unique water districts as potential partners. These game theoretic methods also typically do not consider network interactions between partners, which is critical to characterizing positive synergies and negative interactions that can impact dynamics in the coupled infrastructure network. Lastly, both benefit-cost analysis and game theoretic cost-apportioning methods struggle to represent multiple conflicting objectives. Our novel approach to partnership design that combines detailed disaggregated water system modeling with multiobjective intelligent search seeks to overcome these challenges that have often thwarted traditional partnership design processes.

Line 74: “To do this, we develop a novel approach that uses detailed ensemble modeling of an interconnected water supply network under uncertainty, combined with multiobjective intelligent search to discover the optimal tradeoffs in infrastructure partnership design (Supplementary Fig. S1).”

Reviewer comment 2.7: do you consider operational rules?

This question can be interpreted in two ways. We will answer both in sequence. The first interpretation is “are operational rules represented in the CALFEWS simulation model”? Yes, CALFEWS represents a wide range of operational rules, including but not limited to (a) reservoir flood control, minimum flow, and water supply operations, (b) water rights and interbasin water supply contract operational rules, (c) storage and conveyance infrastructure priority access rules, and (d) groundwater banking operational rules. For existing infrastructure, these operational rules are taken from technical reports and other documentation of the water supply system (see (Zeff et al., 2021)). For candidate new infrastructure investments, these rules are adopted from existing technical reports on the project alternatives where such information is available, and extrapolated from similar existing or planned infrastructure reports when necessary (see (Hamilton et al., 2022)).

The second interpretation of Reviewer comment 2.7 is “are operational rules optimized as part of the intelligent search procedure”? The answer to this is no – our focus was on optimizing the partnership design, a challenging problem in its own right. We do believe that co-optimization of partnerships and infrastructure operational rules would be a very interesting extension of this work, but this is beyond the scope of the current study. This would add significant computational expense, although recent advances in multiobjective reinforcement learning could make this a possibility (Bertoni et al., 2020; Giuliani et al., 2021).

Line 84: “Our findings also emphasize the importance of accounting for the substantial uncertainty in infrastructure performance that emerges due to California’s extreme hydroclimatic variability. Uncertainty in overall infrastructure performance is accompanied by high degrees of heterogeneity in the water supply benefits and financial risks experienced by individual water providers under different partnerships.”

Reviewer comment 2.8: This is a great point, but one that has been made before in water planning context, for example References 1 & 2 below. As it stands, the paragraph reads as ambiguous. I believe here you can introduce more explicitly some of the findings you put forward in the discussion. I think if you briefly summarize the 4 points you describe in the discussion this would make the reader appreciate more the degree of innovation of your work.

Thanks for the helpful references, we have added them to the citations here:

Line 414: However, this will require updated planning frameworks that move beyond expected value-based benefit-cost analyses (e.g., exploratory modeling and robust decision-making approaches (Groves et al., 2014; Kasprzyk et al., 2013; Lempert, 2002; Moallemi et al., 2020; Molina-Perez et al., 2019)), as well as detailed water supply models that can account for complex local-to-regional scale dynamics in coupled hydrologic, infrastructural, and institutional systems under diverse conditions (Zeff et al., 2021). Flexible planning frameworks such as Engineering Options Analysis, Dynamic Adaptive Policy Pathways, and adaptive contract structures can also help decision makers to design for flexibility and delay expensive irreversible investment decisions until they have gathered more information about likely future conditions (de Neufville et al., 2019; Fletcher et al., 2017; Gorelick et al., 2023; Haasnoot et al., 2013; Herman et al., 2020).

This paragraph at the end of Introduction does largely summarize the key findings and implications made in the Discussion:

Lines 84: We find a wide diversity of alternative infrastructure partnerships that exhibit strong tradeoffs between the size of a partnership and partner-level financial risk, and between water supply benefits for the investing partners and the rest of the region. Our findings also emphasize the importance of accounting for the substantial uncertainty in infrastructure performance that emerges due to California’s extreme hydroclimatic variability. Uncertainty in overall infrastructure performance is accompanied by high degrees of heterogeneity in the water supply benefits and financial risks experienced by individual water providers under different partnerships. Lastly, we demonstrate the advantages of our multiobjective intelligent search framework through a baseline comparison focused on the ongoing efforts to rehabilitate the Friant-Kern Canal. As an example, one of the candidate partnership designs discovered using our

intelligent search procedure achieves 58% higher water supply gains while also significantly lowering the risk of extreme cost burdens for investing water providers when compared to the existing status quo canal expansion partnership. These insights are only attainable by moving beyond traditional ad hoc planning practices, expected value benefit-cost frameworks that do not resolve key differences in partners' individual water supply and risk tradeoffs, and highly aggregated regional water supply models that fail to resolve the institutional complexities and local-scale dynamics that drive consequential differences in partners' investment benefits and risks. Our results have important policy implications for California and broader insights for other regions working to develop collaborative investment partnerships to increase the resilience of their water supply systems.

Line 91: "achieves 67% higher water supply gains while also significantly lowering the risk of extreme cost burdens for investing water providers when compared to the existing status quo canal expansion partnership."

Reviewer comment 2.9: Do you consider potential legal constraints?

Please see our response to Reviewer comments 2.10, 3.3, and 1.3.

Line 95: "highly aggregated regional water supply models that fail to resolve the institutional complexities and local-scale dynamics"

Reviewer comment 2.10: what are institutional complexities?

By Institutions, we refer to the wholistic set of human rules and norms impacting the behavior of California's water supply system. The quoted sentence refers to a suite of complex institutional behaviors and constraints that are modeled in CALFEWS, including water rights, interbasin transfer contracts, environmental minimum flow regulations, flood control operations, groundwater banking contracts, and financial risk related to infrastructure partnerships. Please see (Hamilton et al., 2022; Zeff et al., 2021) for a broader discussion of these institutional complexities.

Line 97: "results have important policy implications for California and broader insights for other regions working to develop collaborative investment partnerships to increase the resilience of their water supply systems."

Reviewer comment 2.11: Mention these implications. I think you do a great job at describing these throughout the paper. However, these sentences leave the reader looking for the actual recommendations, many readers may stop reading the paper at this point, so being explicit about this can benefit the paper.

Please see our response to Reviewer comment 2.8. This paragraph already does largely summarize the same policy implications and discussion points that are discussed in more detail later.

Line 135: "The high-fidelity system representation of the model allows us to generate new insights into infrastructure partnership design compared to existing lower-fidelity water supply planning models that cannot resolve the multi-timescale dynamics of managed aquifer recharge or the multi-spatial scale distribution of water supply benefits and financial risks for individual water providers^{23,33}."

Reviewer comment 2.12: Can you provide a few examples here?

One example that showcases these ideas is groundwater banking. To model the future benefits of a groundwater banking investment, we need a model that uses a daily or subdaily time step in order to capture the fast timescale dynamics of peak flow behavior during atmospheric river events in California. Models that use longer weekly or monthly timesteps (which is common for planning studies) cannot accurately capture the impact of infrastructure constraints in a fast-changing high-flow event. At the same time, to assess the value of groundwater banking, one needs to run sufficiently long simulations to capture the impacts of internal climate variability, climate change, and groundwater level change. In the spatial domain, models which do not resolve different water districts as individual nodes with unique water rights portfolios, infrastructure access rights, cropping patterns, and strategic behavior may be unable to characterize the overall coupled system dynamics or the local-scale value tradeoffs of investing in a new collaborative groundwater banking project.

We discuss these topics in the following excerpt from the Introduction, among other places.

Lines 49: Modern water supply systems are governed by a complex set of hydrologic, infrastructural, and institutional drivers and uncertainties (Escriva-Bou et al., 2020; Trindade et al., 2020; Zeff et al., 2021). This presents a significant modeling challenge for water providers trying to estimate the expected benefits and costs of expensive infrastructure investments. For example, understanding the value of groundwater recharge facilities requires detailed modeling of short-timescale dynamics (e.g., atmospheric rivers or snowmelt-driven flood events), long-timescale trends (e.g., climate change and aquifer depletion), and time-evolving infrastructure constraints, water rights, and regulations (Alam et al., 2020; Dillon et al., 2019; Hanak et al., 2018; Kocis & Dahlke, 2017). Moreover, water supply systems in many regions are increasingly interconnected, which can lead to complex emergent behaviors within the coupled network because each water providers' future supply reliability depends not only on its own actions and capabilities, but also on the actions and capabilities of other water providers in the network (Gold et al., 2022; Srikrishnan et al., 2022; Yoon et al., 2022).

Line 142: "This allows us to explore the impact of California's severe internal hydroclimatic variability on the water supply benefits and financial risks resulting from these infrastructure investments^{51,52}."

Reviewer comment 2.13: How this is compared to observed hydroclimatic history?

Please see our response to Reviewer comment 2.1.

Line 144: "The synthetic streamflows used in this study serve two major technical benefits: (1) they permit a well-founded statistical representation of the plausible flood-drought regimes that regional infrastructure investments will face in the near-term and (2) they are intentionally strongly optimistic in neglecting longer-term climate changes to show that even without climate change effects, there are immense uncertainties that challenge our understanding of investment partnerships' water supply and financial risk tradeoffs."

Reviewer comment 2.14: Replace "will face" with "can potentially face"

Thanks, done.

Reviewer comment 2.15: I don't think this is a sound methodological decision. Why would you bias the analysis with optimism? Moreover, most vulnerability assessments in the region are already using downscaled climate series of CMIP6 or CMIP5 ensembles. These ensembles show that precipitation can increase or decrease as compared to historical conditions and this can really impact the level of vulnerability in the zone you are studying. Would it be too difficult to add climate change to the analysis? This addition will make analysis more sound.

Please see our response to Reviewer comment 2.1 above.

Line 155: "The elements of partnership design that are optimized can be described with three questions: (1) In which project does the partnership invest (canal expansion, groundwater bank, or both)? (2) Which subset of 42 water providers in the region should participate? (3) What "ownership share" should be assigned to each partner, governing its capacity in the project and its share of annual debt payment obligations?"

Reviewer comment 2.16: "what happens if they do not participate? how participation is being decided over?"

Our multiobjective intelligent search process is meant to discover an initial suite of candidate partnerships which would be high-performing in principle, but we cannot guarantee that each of these partnerships would be feasible in practice or that each provider will necessarily be interested in participating. There are bound to be legal, fiscal, and personal constraints that are not captured in our model that would impede some partnerships from working in practice. We view our results not as providing "the answer" of what should be built, but rather as a first step in what would need to be an iterative process of learning and design with stakeholder involvement.

The Discussion has been updated to reflect these concerns:

Lines 545: Our results highlight the significant regret associated with traditional ad hoc partnership planning processes and the substantial improvements that can be achieved by combining detailed ensemble water supply modeling with multiobjective intelligent search. However, in practice, partnerships emerge as part of a complex human process within a broader historical and institutional context. Future work should investigate how these computational tools can best be integrated with more traditional stakeholder-based collaborative planning efforts in order to discover new alternatives, illuminate tradeoffs, and improve transparency to break down historical silos (Basdekas & Hayslett, 2021; Moallemi et al., 2021; Smith et al., 2019, 2022). Local planners and operators have important knowledge about the infrastructural and institutional behaviors, as well as legal, fiscal, and personal constraints, that may not be presently captured in our modeling framework. Thus, in practice it would be critical to complement the multiobjective intelligent search with iterative processes of evaluation and human feedback to better capture this explicit and implicit knowledge.

Line 165 (original manuscript): "The vast majority of optimal tradeoff partnerships (99%) were found to invest in both canal expansion and groundwater banking (Fig. 2a)."

Line 165 (edited to reflect updated results): "The majority of optimal tradeoff partnerships (83%) are found to invest in both canal expansion and groundwater banking (Fig. 2a)."

Reviewer comment 2.17: Under which conditions these are not complementary projects?

Please refer to our answer to Reviewer comment 2.20b for more information on the challenge of providing this type of information.

Line 176: “There is also diversity in the concentration of ownership across the partnerships. The smallest partnerships tend to distribute ownership roughly equally (e.g., three partners with shares of ~33% each). Larger partnerships tend to have a few partners with disproportionately large ownership shares and more partners with rather small shares (Fig. 2b, Supplementary Fig. S2).”

Reviewer comment 2.18 (edited to combine two sequential comments from PDF): What do you mean? can you elaborate on the descriptive statistics that support these claims? for example, X% of large partnerships have on average ownership of Y%.

Thanks for this comment, but we removed this claim from the revised manuscript. In our updated results, the smallest partnership is 11 rather than 3 partners, and the trend related to ownership concentration of the smallest partnerships was eliminated as a result.

Line 180: “Considering the geographic context of these partnerships (Figs. 2c-e), we find that certain providers almost always participate and generally carry large ownership shares (black), while others rarely participate and tend to take rather small shares (yellow). There are also providers that regularly participate with small shares (light brown) or that irregularly participate with large shares (teal). These results demonstrate the complexity of partnership design and the range of roles that different water providers can play in partnership creation given their unique contexts: the local infrastructure networks, water rights, hydrologic context, and other factors that impact their ability to procure and store additional surface water as a result of the collaborative investment. Moreover, many providers can span a range of ownership shares across the optimal tradeoff set depending on how the rest of the partnership is constructed (Fig. 2d), which highlights the importance of capturing local-scale dynamics and network interactions between water providers when designing collaborative infrastructure investments.”

Reviewer comment 2.19: In its present form, 2c does not really provide any geographic insights. Is there any key geographic distinction as to why a provider decides to participate or not?

Thank you for this suggestion. The new version of this figure from our updated experiment does display a more consistent spatial trend, and we have expanded the text below to provide more geographic insight.

Lines 199: Considering the geographic context of these partnerships (Figs. 2c-e), we find that certain providers almost always participate and generally carry large ownership shares (dark grey), while others rarely participate and tend to take rather small shares (yellow/light green/light brown). There are also providers that regularly participate with small shares (medium brown) or that irregularly participate with large shares (teal). The set of providers that participate in over half of optimal tradeoff partnerships and that have a median ownership share above 4% (top right quadrant of Fig. 2e) are heavily concentrated in the central region along the Friant-Kern Canal in the vicinity of the Tule River (Figs. 1, 2c). These are the providers most heavily impacted by the heavy subsidence in this area and the subsequent restriction in canal conveyance capacity. Many of these providers are Friant Contractors, but there are also several Friant Contractors that participate infrequently and/or with small shares, as well as several non-

Friant districts that can play an important role in investment partnerships, as we will see in subsequent sections.

Please refer to our response to Reviewer comment 2.20b regarding the challenges of producing a more statistically informed suite of design insights given the sheer size of the design space for this challenging multiobjective problem.

Reviewer comment 2.20a (responding first to highlighted portion) This is all very illustrative, but why? could you run a statistical learning model to describe parametrically the different partnership designs? for example, providers that almost certainly participate and have large ownership shares occur in under which temperature and climate conditions?, under which interest rate conditions?, etc. I think this can be vastly informative for providers because this would provide evidence as to how their own characteristics determine their level of participation.

Thank you for these questions. Before addressing the question about statistical learning, we will address the highlighted portion of the question with two important clarifications about our experimental design. First, our multiobjective intelligent search process selects for partnerships which perform well in aggregate across a range of hydrologic conditions. This results from our objective formulations which aggregate both across time (e.g., average annual captured water gain across a 30-year simulation) and across an ensemble of hydrologic scenarios (e.g., the 90th percentile of cost across 21 scenarios in the optimization stage or 79 scenarios in the reevaluation stage). Our framework thus selects partnerships that are high performing and robust across a full ensemble of streamflow scenarios, rather than selecting a different optimal partnership for each different scenario. This is a subtle but important distinction and means that we can't say which providers "participate and have large ownership shares...under which temperature and climate conditions", because that would presuppose that decisionmakers have knowledge of the future conditions before making their decision of whether to participate or not, which is not realistic. Decision-makers must make long-term planning decisions under uncertainty, without having perfect foresight about the future climate conditions that they will experience.

Second, this experiment only explicitly considers hydrologic aspects of uncertainty via streamflow scenarios. We do not explicitly model changes in temperature or other climatic variables, except insofar as they are implicitly embedded in our synthetic hydrologic scenarios and in the ensemble of downscaled CMIP5 scenarios that we tested. We also do not address other sources of uncertainty related to interest rates or other economic factors. Studying these other sources of uncertainty is important in our opinion, but is beyond the scope of this paper. We are continuing to investigate these issues in ongoing work, as noted in the Discussion:

Lines 531: In this study, we explore hydrologic uncertainty using a broad ensemble sampling of plausible 30-year daily streamflow sequences from a synthetic generator trained on historical observations. This represents an actively optimistic framing because we only account for stationary hydroclimatic variability while neglecting nonstationary climate change. Despite the optimistic framing, we nevertheless find that uncertainties in assessing water supply benefits and financial risks are highly consequential as a result of California's highly variable climate. We also find that our synthetically sampled streamflow scenarios span a similarly wide range of extremes as a multi-model downscaled CMIP5 ensemble for the 2021-2050 period, and that simulated partnership performance across the two ensembles is similar. Future work will consider the impact of a wider range of plausible nonstationary climate futures to account for

the impact of climate change on uncertainty and decision-making. Additionally, it will consider other uncertainties related to future water demand, reservoir operations, infrastructure performance, and cost. These factors are expected to further widen the envelope of uncertainty around aggregate and partner-level impacts and exacerbate the challenge of designing robust partnerships.

Reviewer comment 2.20b (responding to highlighted portion): This is all very illustrative, but why? Could you run a statistical learning model to describe parametrically the different partnership designs? For example, providers that almost certainly participate and have large ownership shares occur in under which temperature and climate conditions?, under which interest rate conditions?, etc. I think this can be vastly informative for providers because this would provide evidence as to how their own characteristics determine their level of participation.

We now return to the Reviewer's question about whether a statistical model or scenario discovery approach can be designed to learn which water provider characteristics lead to higher participation in infrastructure partnerships and/or better performance in those partnerships.

The partnership design experiment in this paper consists of 40 candidate water provider partners, each with its own unique combinations of water rights, interbasin transfer project contracts, locations, infrastructure connections, neighbors, groundwater banking arrangements, local groundwater pumping and recharge facilities, municipal water use and cropping demand patterns, and other characteristics which impact system dynamics and performance. Overall, this amounts to thousands of degrees of freedom for the partnership design problem, which means it is very challenging to attribute any differences across partnerships to individual partner-level characteristics. Our Pareto set only has 270 unique partnerships from which to learn patterns, which increases the risk of overfitting. Moreover, the Pareto set is a learned manifold within the broader design space, and thus represents a highly non-random sample which presents a challenge for learning generalizable relationships using statistical or machine learning approaches.

Moreover, each providers' benefits may be highly nonlinear with respect to their ownership share in the project due to constraints in their infrastructure, water rights portfolio, or the non-aligned timing of winter peak flows and summer peak irrigation demands. An additional challenge is the important interactions between different partners, with certain pairs or higher order groupings of providers potentially having significant negative or positive interactive effects based on their unique patterns of use in the constrained coupled infrastructure system.

Overall, given the expansive design space, heterogeneous partner-level characteristics, and higher order interactions at play, this is an underdetermined problem for our current dataset which is unlikely to yield generalizable results without overfitting in our opinion. Although we have tried several approaches to statistical learning or clustering with this dataset, these techniques were not found to produce valuable generalizable insights in our experience thus far. The sheer level of complexity and heterogeneity in partner-level characteristics makes this a particularly challenging problem.

While we agree that it would be valuable to learn generalizable characteristics that lead partners to participate in high-performing partnership, this type of scenario discovery would require a significantly new design of experiments based on largescale randomized sampling, with very substantial computational demands. This is beyond the scope of the present study but we hope to investigate these issues in a follow up study.

Reviewer comment 2.21 (referring to last sentence in text above): I agree, but which type of network interactions determines the outcome?

Thank you for this question. We have expanded the text to provide more context on the important network interactions at play.

Line 211: These results demonstrate the complexity of partnership design and the range of roles that different water providers can play in partnership creation given their unique contexts: the local infrastructure networks, water rights, hydrologic context, and other factors that impact their ability to procure and store additional surface water as a result of the collaborative investment. Moreover, many providers can span a range of ownership shares across the optimal tradeoff set depending on how the rest of the partnership is constructed (Fig. 2d). This results from the large number of provider interactions across the coupled statewide-to-local scale infrastructure network (e.g., shared canal space) as well as interactions in the institutional space (e.g., water contracts and groundwater banking arrangements). These interactions can lead to path-dependent system dynamics across the region, where each providers' water supply operations can impact the availability of water and infrastructure capacity for the other providers in the network. This highlights the importance of capturing local-scale dynamics and network interactions between water providers when designing collaborative infrastructure investments (Hamilton et al., 2022; Zeff et al., 2021).

Line 204: "For context, 100 GL (81 kAF) is equivalent to 15% of Lake Millerton's capacity or 4% of Kern County's average irrigated acreage."

Reviewer comment 2.22: This is useful but I think the level of scale is missing. how big the demand of Kern's County's with respect to overall demand in San Joaquin? if it is too small, maybe that 4% improvement is not very relevant, if it quite substantial, this 4% is extremely relevant, please clarify.

Thanks, we have improved the context provided in the text below:

Line 236: The optimal tradeoff partnerships can increase average surface water deliveries to partners by anywhere from 48 to 85 GL/year. For context, 85 GL (69 kAF) is equivalent to 13% of Millerton Lake's capacity, or 3.4% of the average annual groundwater overdraft for the entire San Joaquin Valley (Hanak et al., 2017). This represents a meaningful increase in surface water deliveries at the local scale which could help partners reduce their over-reliance on groundwater pumping in order to meet their obligations under the Sustainable Groundwater Management Act, although this would need to be only be one piece in a larger portfolio of water supply resiliency investments to address the scale of the challenge of groundwater overdraft in the Valley (Hanak et al., 2020).

Line 208: "the effect on groundwater pumping varies widely across the optimal tradeoff partnerships, from a reduction of 74 GL/year to an increase of 3 GL/year on average."

Reviewer comment 2.23: Please provide context, how big are these reductions as compared to the Sustainable Groundwater Act targets?

Although this particular statement has been removed in the revised manuscript due to our problem reformulation that removed groundwater use as an objective, we have nonetheless provided more context on SGMA in the response above to Reviewer comment 2.22.

Line 226: Figure 3

Reviewer comment 2.24: This graph presents a very interesting result, however the graph is only referenced via parenthesis in the main body of the text.

There are four full paragraphs of discussion of Figure 3 (Lines 235-291). Each of these paragraphs cites Fig. 3 parenthetically when referring to specific results, as is standard.

Reviewer comment 2.25: The graph is showing that the higher the number of partners in the partnership the higher the probability that a subset of these will face financial risk, is this correct? if so, why adding more partners ends up exposing them to financial risk? what rule in the partnership is creating this risk, are all partners financing the projects in the same proportion, regardless of their size or capacities? That is a policy relevant fact we would like to know.

Yes, we find that as the partnership size grows beyond a certain point (e.g., 20+ partners), it becomes increasingly likely that a subset of these partners will experience poor water supply benefits and subsequent financial risk. This is consequence of the conjunction operation inherent to the maximin formulation for the financial risk objective (minimizing cost for the worst-off partner). This is somewhat analogous to adding a new objective for each new partner; the number of tradeoffs can grow factorially with the number of objectives in a multiobjective problem, making it rapidly more difficult to satisfy on all objectives at once (Coello Coello et al., 2007; Giuliani et al., 2016; Woodruff et al., 2013). In simple terms: the more partners you have, the harder it is design the partnership perfectly such that each partners' water supply benefits are in line with its ownership share and cost burden.

Regarding the second question, it is important to note that for the partnerships in our optimal tradeoff set, the partners are not generally financing the projects in equal proportion. Rather, each partner's financial contribution is set by its ownership share, which is a decision variable in the multiobjective search. Rather than assuming the cost shares a priori, we allow the search algorithm to provide evolutionary pressure that iteratively improves the distribution of ownership shares across a partnership such that the ownership shares (and resulting cost shares) are more in line with each partners' ability to capture surplus water (as a function of its water demand, water rights portfolio, local infrastructure capacity, etc.). This is explained as follows:

Line 163: The elements of partnership design that are optimized can be described with three questions: (1) In which project does the partnership invest (canal expansion, groundwater bank, or both)? (2) Which subset of 40 water providers in the region should participate? (3) What "ownership share" should be assigned to each partner, governing its capacity in the project and its share of annual debt payment obligations? Additional details on our partnership design and evaluation framework can be found in Methods and Supplementary Notes S2-3.

Line 603: The elements of partnership design to be optimized can be understood as answering three questions. First, which infrastructure project should be built: canal expansion, groundwater banking, or both? Second, out of 40 candidate water providers in the region (Fig. 1), which subset should work together as partners? Third, once the set of partners has been

established, how should the “ownership shares” be distributed amongst them? Each partner’s ownership share dictates its share of priority capacity in the new infrastructure as well as its share of annual debt payment obligations, following Hamilton et al. (2022).

Line 365: The partners within the Compromise Partnership experience a range of expected costs of gains for their water supply benefits (Fig. 4c), which is common across the optimal tradeoff partnerships. The heterogeneity of expected costs stems from the similarly heterogeneous expected captured water gains at the partner level (Supplementary Fig. S8a). The latter is not inherently problematic so long as each partner’s captured water gain is appropriately matched to its ownership share in the project and thus its share of the annual debt payments. For example, if Provider A receives twice as much captured water gain as Provider B, but also makes annual debt payments that are twice as large, then the two providers are effectively paying the same unit cost for their captured water gains. However, we find the ownership shares to be imperfectly matched in this and other partnerships, so that some providers pay more than their “fair share” on average and others pay less (Fig. 4c).

Line 477: Second, although the Compromise Partnership and the Status Quo Partnership have significant overlap in participating providers (Figs. 4a & 5a), the Compromise Partnership discovered through multiobjective intelligent search removes several Friant Contractors with marginal benefits and significant financial risks. It also adds several other non-Friant providers that stand to benefit from the infrastructure investment. Widening the net beyond the Friant Contractors allows for a more diversified portfolio of water supplies and demands and increases the utilization of the infrastructure across a range of seasons and conditions. Lastly, the ownership shares and annual debt payments as refined through the multiobjective intelligent search process are better matched to partner-level captured water gains in the Compromise Partnership than the Status Quo Partnership, which helps to equalize the cost of gains across partners. These results highlight the significant regret associated with traditional ad hoc partnership design processes and the substantial improvements that can be achieved by combining detailed ensemble water supply modeling with multiobjective intelligent search.

Additional mathematical detail on the decision variable formulation that defines the procedure for mapping each candidate numeric decision vector proposed by the MOEA into a set of partners and their relative ownership shares in the infrastructure investment is provided in Supplementary Note S3.

Reviewer comment 2.26: Minor comment: I think the first parrallel is not useful as you are using the color legend to depict that same dimension.

Thanks, but we respectfully disagree. This redundancy is helpful for easing interpretation and is very common across a range of prior publications that use parallel coordinate plots. Including each metric as a parallel axis also makes explicit the relationship between axes 1 & 2 via diagonal lines, which would be less obvious if the first objective were only represented by color. In our opinion there is not much to gain by reducing from 4 to 3 vertical axes, so we prefer to keep it as is.

Line 232: “In general, the partnerships that achieve the largest surface water gains and groundwater pumping reductions tend to have many partners (Fig. 3, Supplementary Figs. S3-S4).”

Reviewer comment 2.27: I don't think the graph is showing this. To clarify I recommend you report descriptive statistics, for example, partnerships with more than 20 partners achieve on average GL/yr of

captured water gain. Also, I think it would be valuable to comment on the partnerships that achieve the highest gains? are they all the same size? or maybe these are heterogeneous enough that they do not display common attributes?

Thanks for this suggestion but, our updated results do not show this particular trend strongly and thus we have removed this claim from the paper.

However, in the spirit of this comment we have added this type of detailed quantification in several other places in the text, for example:

Line 269: For example, over half of partnerships with 24+ partners have at least one partner paying over \$1000/ML, while each partnership in which all partners pay under \$200/ML have 16 or fewer partners (Supplementary Fig. S2-S3).

Line 244: "However, it is also critical to understand the significant financial risks that can arise from these arrangements and to ensure that all partners are receiving adequate benefits to justify their debt."

Reviewer comment 2.28: This is a major finding of the study, can you elaborate more on why this happens? is it because all partners invest in the same proportion?

Please see our response to Reviewer comment 2.25.

Line 265: "Excluding partnership designs with significant negative impacts on non-partners from the optimal tradeoff set to account for potential political and legal constraints causes the best achievable captured water gains and pumping reductions to be reduced by 16% and 6%, respectively (Supplementary Fig. S6). This negative interaction represents a major challenge for supply reliability and groundwater sustainability efforts and points to the need for more coordinated regional infrastructure planning."

Reviewer comment 2.29 (referencing first sentence from Line 265 quote above): This is a very interesting result. However, I think it will make your argument stronger if you are more specific about this claim. For example, could you clarify what do you mean by significant negative impacts? For example a significant impact is one that is higher than the mean? or perhaps higher than Q75? It would be very useful to know the concrete threshold that makes a negative impact significant.

Thank you for pointing this out as unclear. These results were generated by screening out all partnerships for which the expected captured water gain for non-partners metric has a negative value. The text has now been updated to more clearly convey this.

Line 308: When we exclude partnership designs from our optimal tradeoff set which are expected to reduce total average water deliveries to non-partners to account for potential political and legal constraints, the best achievable captured water gains for partners is reduced by 6% (Supplementary Fig. S5). Excluding partnerships that would cause injury to individual providers (rather than in aggregate) or in individual years or hydrologic scenarios (instead of expected value across many scenarios/years) would be expected to substantially reduce the set of feasible partnerships further.

Supplementary Fig. S5 caption also helps to explain the screening procedure used to generate this result.

Supplementary Fig. S5: Parallel coordinate plot excluding partnerships with negative expected external impacts. All partnerships that reduce the total average water deliveries to non-partners in the region are screened out, as represented by the grey bar.

Reviewer comment 2.30 (referencing second sentence from Line 265 quote above): Could you be more specific about this? So do we need better coordination schemes? or better compensation schemes? or better markets? or all of the above?

Thanks, good question. We have expanded on this as follows:

Line 314: This negative interaction represents a major challenge for supply reliability and groundwater sustainability efforts and points to the need for more coordinated regional infrastructure planning efforts to develop synergistic water supply portfolios with regional benefits and minimal third party injuries, as well as programs to mitigate and compensate for these injuries.

Line 277: "Moreover, the striking performance tradeoffs that emerge from nuanced changes in partner selection and ownership distribution highlight the significant advantages of pairing detailed water supply models like CALFEWS (capable of resolving daily-timescale coupled hydrologic, infrastructural, and institutional dynamics at the scale of individual water providers) with multiobjective intelligent search algorithms like the Borg MOEA (capable of exploring a much wider range of candidate partnership designs compared to traditional ad hoc planning processes)."

Reviewer comment 2.31: I find this argument key to the paper, I wonder if it would be better to mention this in the introduction.

Thanks, but this is already highlighted in the last paragraph of the Introduction.

Line 293: "The captured water gain and pumping reduction metrics in Fig. 3 are calculated as the mean values across the 64 sampled 30-year daily streamflow sequences, while the worst-partner cost of gains metric is calculated using the 90th percentile across scenarios."

Reviewer comment 2.32: This statistical detail is what I recommend adding to the paragraphs when a major result is put forward.

Thanks for this suggestion, but we prefer to leave the detailed discussion of uncertainty for this section rather than bring it forward. The aggregation methods are mentioned briefly in the preceding section as necessary to expand the initial results. But we feel that digging deeper into the details of ensemble statistics at the beginning of the Results is likely to confuse some readers and water down the narrative arc that we developed. Our storyline intentionally begins with aggregated results (e.g., expected value), which are the standard in many planning studies, and then moves from there into ensemble results in the present section to highlight the impact of accounting for uncertainty. We feel that this presentation increases the impact of our results.

Line 301: "The Compromise Partnership has 17 partners investing in both canal expansion and groundwater banking (Fig. 4a) and is selected for its relatively high performance across all five performance objectives (see Methods)."

Reviewer comment 2.33: This is ambiguous, please clarify, what is "relative high performance"?

The selection process is described in detail in the Methods (Section "Selection of Status Quo & Compromise Partnerships").

Line 728: By comparison, the Compromise Partnership is selected as a representative solution from the broader 270-member optimal tradeoff set that outperforms the Status Quo Partnership and performs relatively well across all four objectives. This partnership is selected through a two-step process (Supplementary Fig. S18). First, we find the subset of optimal tradeoff partnerships that perform equal to or better than the Status Quo Partnership on all four aggregate objective metrics. The Compromise Partnership is then selected out of this subset by finding the partnership with the lowest worst-partner cost of gains.

Line 323: "if Provider A receives twice as much captured water gain as Provider B, but also makes annual debt payments that are twice as large, then the two providers are effectively paying the same unit cost for their captured water gains. However, we find the ownership shares to be imperfectly matched in this and other partnerships, so that some providers pay more than their "fair share" on average and others pay less (Fig. 4c)."

Reviewer comment 2.34: This is extremely useful. Note, however, that I don't we can say we face the same level of heterogeneity across all metrics. For example, for "cost of gains for worst-off partner" the variance of the outcome is apparently much less as compared to the other metrics. Is this because the worst-off partner is always the same?

The worst-off partner is not always the same. The rank-order of partners' costs of gains are not necessarily stable across hydrologic scenarios; for example, a hypothetical partnership could have Partner A experiencing the highest cost of gains in 75% of scenarios, Partner B in 15% of scenarios, and Partner C in 10% of scenarios. Our worst-off cost of gains metric is designed to take the worst cost experienced by any partner in a given hydrologic scenario, and thus in this hypothetical it would be equal to Partner A/B/C cost in 75/15/10% of scenarios.

This heterogeneity occurs due to the heterogeneous characteristics of different water providers and their unique responses and vulnerabilities to different types of hydrologic scenarios. For example, hypothetically Partner A could be most vulnerable to scenarios with short-lived but intense dry spells, while Partner B could be most vulnerable to scenarios with moderate but long-lasting droughts, and Partner C could be the weakest performer in relatively wet hydrologic scenarios without any extreme drought periods.

More generally, the Reviewer is correct that there are different levels of uncertainty/variability across the three uncertain objectives. Although the uncertainty range for the cost of gains for the worst-off partner metric may be smaller as a fraction of its axes in Figs. 4-5 compared to the other two metrics, this uncertainty band can still be highly consequential for decision-makers.

Line 357: The worst-partner cost of gains also spans a wide range, from \$105/ML in the best scenario (58% of the 90th percentile metric of \$180/ML) to \$407/ML in the worst scenario (227% of the 90th percentile metric). This cost differential could easily be the difference between a project that affordably improves surface water access and one that provides little water supply benefit and overburdens water providers with debt and their customers with rate increases.

Line 346: "This risk is not evenly distributed, with some partners experiencing a disproportionate share of extreme costs (e.g., Providers 3 and 4) and others experiencing uniformly low costs across the sampled hydrologic scenarios (e.g., Providers 7 and 12)."

Reviewer comment 2.35 (referring specifically to partners experiencing extreme costs): Which percent of partners experience this?

Thanks, we have edited to provide more concrete information:

Line 394: However, the cost of gains for the worst-off partner (red distribution) has a much wider range, reaching over \$400/ML in the most challenging scenario. This risk is not evenly distributed, with some partners experiencing a disproportionate share of extreme costs (e.g., three out of sixteen partners experiencing worst-case costs over \$300/ML) and others experiencing uniformly low costs across the sampled hydrologic scenarios (e.g., four partners experiencing worst-case costs under \$100/ML).

Line 355: "Our expressly optimistic framing of future hydroclimatic conditions in California neglects any nonstationary effects from anthropogenic warming or low-frequency decadal climate oscillations. However, even under such optimistic conditions, the extreme levels of interannual hydroclimatic variability experienced in California lead to significant decision-relevant uncertainty in cooperative infrastructure investment outcomes over the next several decades."

Reviewer comment 2.36: "However, it would be fundamental to stress test the analysis against more adverse climate conditions? I believe it is possible that under high climate stress, the distribution of cost and benefits may be less heterogeneous and it would very important to show that in the context of this work?"

Please see our response to Reviewer comment 2.1.

Line 362: "We caution that decision-makers should carefully consider the impacts of uncertainty and financial risk before committing to significant debt associated with new infrastructure partnerships."

Reviewer comment 2.37: Would it be possible to plan these investments in an adaptive form? that would allow partners to commit to less risky investments in the short-term and expand these investments in the long-term once more evidence about the potential climate trajectory that the region is facing becomes available.

Thanks, good question. In our opinion adaptive pathways and flexible contracts hold great promise here. We have expanded the following discussion:

Line 412: We caution that decision-makers should carefully consider the impacts of uncertainty and financial risk before committing to significant debt associated with new infrastructure

partnerships. However, this will require updated planning frameworks that move beyond expected value-based benefit-cost analyses (e.g., exploratory modeling and robust decision-making approaches (Groves et al., 2014; Kasprzyk et al., 2013; Lempert, 2002; Moallemi et al., 2020; Molina-Perez et al., 2019)), as well as detailed water supply models that can account for complex local-to-regional scale dynamics in coupled hydrologic, infrastructural, and institutional systems under diverse conditions (Zeff et al., 2021). Flexible planning frameworks such as Engineering Options Analysis, Dynamic Adaptive Policy Pathways, and adaptive contract structures can also help decision makers to design for flexibility and delay expensive irreversible investment decisions until they have gathered more information about likely future conditions (de Neufville et al., 2019; Fletcher et al., 2017; Gorelick et al., 2023; Haasnoot et al., 2013; Herman et al., 2020).

Reviewer #2 references

1. *Developing a Robust Water Strategy for Monterrey, Mexico: Diversification and Adaptation for Coping with Climate, Economic, and Technological Uncertainties | RAND*
2. *Developing Robust Strategies for Climate Change and Other Risks: A Water Utility Framework on JSTOR*

Reviewer #3 (Remarks to the Author):

This study illustrated the importance of water infrastructure partnership design in terms of water gains to partners and non-partners, groundwater pumping, and financial risk. It is a well-written manuscript that points out an important yet under-researched consideration in water resources management.

Thank you to Reviewer #3 for taking the time to review our paper. We are glad that you found it clear and insightful. Please see below for our responses to your comments.

There are a few questions that came up in my reading that I think would improve the manuscript if addressed.

Reviewer comment 3.1: First, I understand a main goal is to show the differences between the robust, multi-objective decision problem formulation vs a more aggregated but traditional benefit-cost analysis, but it still seems strange that an estimate of the central tendency of cost of water gains is not included as a decision metric in the MOEA. If this is because it is too similar to other metrics (e.g., captured water gain), that would be helpful for the reader to understand.

Thanks, this is a good question. As you suggest, the reason is that the expected captured water gain metric conveys very similar information. The expected cost of water gains is defined as the cost of the infrastructure project divided by the expected captured water gain. Thus there is a monotonic one-to-one map between the two (a $1/x$ relationship). Although the expected cost of water gains can differ by a scalar multiplier when two partnerships opt for different projects (e.g., comparing the cost of canal-only vs canal+bank), in initial testing this was not found to provide sufficient differentiation in terms of unique

evolutionary pressure to warrant the computational cost of including it as an additional objective in the multiobjective search. We have added the following sentence to the Methods:

Line 676: Objective Metric 4 is calculated using the 90th percentile across sampled hydrologic scenarios and the max (i.e., worst-case) across partners in order to favor partnerships that can provide robust affordable water supply benefits across a wide range of plausible hydrologic conditions (McPhail et al., 2018; Quinn et al., 2017). Note that we do not include the expected value of the cost of gains as an independent metric because it is monotonically related to the expected captured water gains metric for a given infrastructure investment and therefore it would not introduce significant independent information or tradeoffs to the decision formulation. The 90th percentile worst-partner formulation draws from previous efforts to define robust multiobjective problem formulations in noisy water resources simulation-optimization contexts (Herman et al., 2014; McPhail et al., 2018; Quinn et al., 2017).

Reviewer comment 3.2: Second, it is unclear why the number of partners would need to be maximized, especially since you are already accounting for maximizing the gains for both partners and non-partners. The authors explain this a little at lines 42-45, but it seems unlikely that more partners would always be better across the board. Surely there is some cost to getting all the partners to agree on contracts and other requirements.

This metric is a proxy for the inclusive collaborative goals of California's Water Resilience Portfolio Initiative, which states that collaboration is one of its central goals for developing a more resilient water supply system. Secondly, because the worst-partner cost of gains metric will put downward evolutionary pressure on partnership size during the computational search with the Borg MOEA, this metric is designed to apply the opposite evolutionary pressure so that we can more fully explore the space of possible partnerships within the search phase. It is true that there will be tradeoffs for large partnerships, and that past a certain point the benefits of collaboration will decline. This is consistent with our results and discussion that emphasize the importance of considering tradeoffs and partner-level incentives that reduce the feasibility of the largest partnerships.

The point about additional unquantified costs is a good one, and we have expanded our discussion on the tradeoffs of large partnerships:

Line 276: However, it is also critical to understand the limits of large-scale collaboration and the significant financial risks that can arise from these arrangements, in order to ensure that all partners are receiving adequate benefits to justify their debt. There are also likely additional tradeoffs beyond those quantified in our work, such as costs, delays, and other challenges related to coordinating and contracting across a large and diverse set of partners with their own incentives and historical relationships (Hansen et al., 2020).

Reviewer comment 3.3: Third, some more background into the legal constraints behind water infrastructure partnerships in California would be helpful for the reader to understand what losses to non-partners might be acceptable. The authors say that "more guidance is needed from the state" but

there must be some historical information on lawsuits to help understand what the threshold for legal action would be in practice.

Thanks for this suggestion. The strict threshold in principle is that any change to water rights must cause “no injury” to third parties. In practice, there are many case-specific nuances and different modeling approaches that can affect how injury is defined within a particular dispute. This is handled on a case-by-case basis in the courts and in front of the State Water Resources Control Board, and to our knowledge there does not exist any clear guidance on generalized principles. We have expanded this discussion with additional detail on this complexity and the resulting costs.

Line 292: This raises important questions about the extent to which non-participating water providers have the ability to block infrastructure investments that could negatively impact them. Water providers investing in new infrastructure in California must navigate the state’s complex web of different water rights, environmental laws, and administrative procedures (Escriva-Bou et al., 2020; Hanak et al., 2018). In particular, as in other western states, many changes related to water rights and diversion and storage patterns require that the change will cause “no injury” to other water right holders (Statutory Water Rights Law and Related California Code Sections, 2017). Third parties can object if they believe any injury has occurred, and outcomes are adjudicated in court or with the State Water Resources Control Board on a case-by-case basis based on the unique hydrologic and legal context. Paying for hydrologic modeling studies and legal analysis to support the validity of any changes to water diversion or storage patterns, as well as potentially compensating any injured parties, can add significant expense and delays to the approval process (Szeptycki, Leon F. et al., 2015). For example, previous studies have found that transaction costs related to hydrologic modeling and legal support required for approval of water right transfers in California and Colorado can increase the total cost of the transfer by 100% or more (Hagerty, 2023; Womble & Hanemann, 2020). In California, the diversity of legal regimes and water laws in the state (e.g., prior appropriative rights vs. riparian rights vs. federal/state water supply contracts) further complicates the evaluation of third party injury related to alternative water supply options. More guidance is needed from the state to streamline procedures and facilitate more collaborative partnerships. When we exclude partnership designs from our optimal tradeoff set which are expected to reduce total average water deliveries to non-partners to account for potential political and legal constraints, the best achievable captured water gains for partners is reduced by 6% (Supplementary Fig. S5). Excluding partnerships that would cause injury to individual providers (rather than in aggregate) or in individual years or hydrologic scenarios (instead of expected value across many scenarios/years) would be expected to substantially reduce the set of feasible partnerships further. This negative interaction represents a major challenge for supply reliability and groundwater sustainability efforts and points to the need for more coordinated regional infrastructure planning efforts to develop synergistic water supply portfolios with regional benefits and minimal third party injuries, as well as programs to mitigate and compensate for these injuries.

Reviewer comment 3.4: Fourth, some of the results seem under-discussed. For example, what characteristics lead to some providers participating and/or carrying large shares of the project cost? Similarly, what characteristics lead to a partner paying more than their "fair share"?

Thanks for this question. While we agree that this type of information would be very valuable, the size and complexity of the design space and heterogeneity of response across water providers means that it is very challenging to produce generalizable answers to these types of questions. This would require a full redesign of the computation experiment and is beyond the core focus of the present paper, but we hope to address it in ongoing work. Please see our response to Reviewer comment 2.20b for more details.

Reviewer comment 3.5: Finally, the authors make a convincing argument for the need for more complex modeling and decision-making to improve water infrastructure partnership design, but it seems unlikely that water utilities would be able to do this on their own (run BORG, for example). And the complexity will be amplified when more than two projects are considered. Should utilities partner with universities to do this? Some more discussion on the feasibility of this argument would be beneficial.

Thanks, good point. We have added the following to the Discussion:

Line 568: Lastly, we acknowledge that the complexity and computational requirements of our multiobjective intelligent search workflow may initially present a challenge for water providers looking to adapt this analysis for new contexts. This will also require breaking down silos across engineering, planning, and policy experts. University researchers, cooperative extension services, and federal/state planning agencies should invest resources towards developing significant capacities for facilitation, training, and technology transfer to enable efficient application of cutting-edge computational tools to critical public planning challenges. The proliferation of free and open-source software and the declining costs of computing resources should also continue to reduce the barriers to this type of analysis over time.

Appendix A

In the process of revising our manuscript based on the Reviewers' comments, we found an error in the CALFEWS model. The error was unrelated to any of the Reviewers' specific comments, but we nevertheless determined that it was consequential enough to require a fix and a rerun of the computational experiment in this paper.

The error related to how groundwater exchange transactions are accounted for in CALFEWS. In California, many water districts have groundwater exchange relationships. These exchanges often involve "paper" water transfers. For example, if District A wants to acquire banked groundwater that belongs to District B, it may compensate District B by transferring some of its surface water allocation for that year.

The District B now has a “paper balance” that gives it the right to request water in the future from a particular surface water contract that originally belonged to the District A. CALFEWS operationalizes several types of surface and groundwater exchanges that occur in California. It does this by tracking paper water transfers which give a district a right to receive some amount of water that it does not originally have a right to receive, based on the groundwater exchange.

We found an error in CALFEWS in which one particular type of groundwater exchange was not properly resetting its paper balances to zero with the start of each water year. The result was that a district could improperly take advantage of its paper balance for many years in a row rather than just one time. This falsely inflated the value of groundwater banking for this type of district because it received more surface water over the course of a simulation than it should have a legal right to do.

After isolating and fixing this problem, we verified that all of the paper balances are resetting to 0 at the beginning of each water year. Figure A1 shows the trajectories of all paper balances across all districts and contract types for our 30-year simulations from GFDL-ESM2M RCP 4.5. Notice how each paper balance accumulates over the course of a water year and is reset at the start of a new water year. This is consistent with the way that CALFEWS expects paper balances to behave and is verification that the error has been resolved.

One important thing to note is that this error was only being activated in a very circumscribed set of circumstances that are not commonly seen in the as-is water system. However, the multiobjective intelligent search was able to exploit this error by designing partnerships in such a way that they would take advantage of this error, thus seeing “free water”. Thus, although we do not expect this to meaningfully change the results of past studies where this exchange mechanism is rare, it was able to significantly change the construction of partnership designs in this work. For this reason, we decided to rerun our full experiment after fixing the error to ensure that it would not improperly bias results.

Figure A1: Paper balances for all water districts and water contracts over the course of a 2021-2050 simulation from downscaled GFDL-ESM2M RCP4.5 simulation. Each balance is found to properly reset to 0 at the start of each water year.

References cited in this response

- Alam, S., Gebremichael, M., Li, R., Dozier, J., & Lettenmaier, D. P. (2020). Can Managed Aquifer Recharge Mitigate the Groundwater Overdraft in California's Central Valley? *Water Resources Research*, 56(8). <https://doi.org/10.1029/2020WR027244>
- Anderies, J. M., Folke, C., Walker, B., & Ostrom, E. (2013). Aligning key concepts for global change policy: Robustness, resilience, and sustainability. *Ecology and Society*, 18(2). <https://doi.org/10.5751/ES-05178-180208>
- Avelino, F. (2021). Theories of power and social change. Power contestations and their implications for research on social change and innovation. *Journal of Political Power*, 14(3), 425–448. <https://doi.org/10.1080/2158379X.2021.1875307>

- Basdekas, L., & Hayslett, R. (2021). *Improving Tradeoff Understanding in Water Resource Planning Using Multi-Objective Search* (9781605735535).
<https://www.waterrf.org/research/projects/improving-tradeoff-understanding-water-resource-planning-using-multi-objective>
- Basheer, M., Nechifor, V., Calzadilla, A., Ringler, C., Hulme, D., & Harou, J. J. (2022). Balancing national economic policy outcomes for sustainable development. *Nature Communications*, *13*(1), 5041–5041. <https://doi.org/10.1038/s41467-022-32415-9>
- Bertoni, F., Giuliani, M., & Castelletti, A. (2020). Integrated Design of Dam Size and Operations via Reinforcement Learning. *Journal of Water Resources Planning and Management*, *146*(4), 1–12. [https://doi.org/10.1061/\(ASCE\)WR.1943-5452.0001182](https://doi.org/10.1061/(ASCE)WR.1943-5452.0001182)
- Bird, R. M., & Slack, E. (2017). Financing Infrastructure: Who Should Pay? *SSRN Electronic Journal*.
<https://doi.org/10.2139/ssrn.3083743>
- Brekke, L. D., Maurer, E. P., Anderson, J. D., Dettinger, M. D., Townsley, E. S., Harrison, A., & Pruitt, T. (2009). Assessing reservoir operations risk under climate change. *Water Resources Research*, *45*(4), 2008WR006941. <https://doi.org/10.1029/2008WR006941>
- Brekke, L., Thrasher, B. L., Maurer, E. P., & Pruitt, T. (2013). *Downscaled CMIP3 and CMIP5 climate projections: Release of downscaled CMIP5 climate projections, comparison with preceding information, and summary of user needs*.
- Ciullo, A., Kwakkel, J. H., De Bruijn, K. M., Doorn, N., & Klijn, F. (2020). Efficient or Fair? Operationalizing Ethical Principles in Flood Risk Management: A Case Study on the Dutch-German Rhine. *Risk Analysis*, *40*(9), 1844–1862. <https://doi.org/10.1111/risa.13527>
- Coello Coello, C. A., Lamont, G. B., & Van Veldhuizen, D. A. (2007). *Evolutionary Algorithms for Solving Multi-Objective Problems* (2nd ed.). Springer Science+Business Media, LLC.
<https://doi.org/10.1046/j.1365-2672.2000.00969.x>

- Cohen, J. S., Zeff, H. B., & Herman, J. D. (2020). Adaptation of Multiobjective Reservoir Operations to Snowpack Decline in the Western United States. *Journal of Water Resources Planning and Management*, 146(12), 04020091–04020091. [https://doi.org/10.1061/\(asce\)wr.1943-5452.0001300](https://doi.org/10.1061/(asce)wr.1943-5452.0001300)
- Collier, M. A., Jeffrey, S. J., Rotstayn, L. D., Wong, K. K.-H., Dravitzki, S. M., Moeseneder, C., Hamalainen, C., Syktus, J. I., Suppiah, R., Antony, J., El Zein, A., & Atif, M. (2011, December 12). The CSIRO-Mk3.6.0 Atmosphere-Ocean GCM: Participation in CMIP5 and data publication. *Chan, F., Marinova, D. and Anderssen, R.S. (Eds) MODSIM2011, 19th International Congress on Modelling and Simulation*. 19th International Congress on Modelling and Simulation. <https://doi.org/10.36334/modsim.2011.F5.collier>
- Cong, S., Nock, D., Qiu, Y. L., & Xing, B. (2022). Unveiling hidden energy poverty using the energy equity gap. *Nature Communications*, 13(1). <https://doi.org/10.1038/s41467-022-30146-5>
- de Neufville, R., Smet, K., Cardin, M.-A., & Ranjbar-Bourani, M. (2019). Engineering Options Analysis (EOA): Applications. In *Decision Making under Deep Uncertainty* (pp. 223–252). Springer International Publishing. https://doi.org/10.1007/978-3-030-05252-2_11
- De Souza, S., Medellín-Azuara, J., Lund, J. R., & Howitt, R. E. (2011). *Beneficiary Pays Analysis of Water Recycling Projects*. http://www.waterboards.ca.gov/water_issues/programs/grants_loans/water_recycling/docs/ec_on_tskfrce/beneficiarypays.pdf
- Dillon, P., Stuyfzand, P., Grischek, T., Lluria, M., Pyne, R. D. G., Jain, R. C., Bear, J., Schwarz, J., Wang, W., Fernandez, E., Stefan, C., Pettenati, M., van der Gun, J., Sprenger, C., Massmann, G., Scanlon, B. R., Xanke, J., Jokela, P., Zheng, Y., ... Sapiano, M. (2019). Sixty years of global progress in managed aquifer recharge. *Hydrogeology Journal*, 27(1), 1–30. <https://doi.org/10.1007/s10040-018-1841->

- Dobbin, K. B., & Lubell, M. (2021). Collaborative Governance and Environmental Justice: Disadvantaged Community Representation in California Sustainable Groundwater Management. *Policy Studies Journal*, 49(2), 562–590. <https://doi.org/10.1111/psj.12375>
- Dufresne, J.-L., Foujols, M.-A., Denvil, S., Caubel, A., Marti, O., Aumont, O., Balkanski, Y., Bekki, S., Bellenger, H., Benshila, R., Bony, S., Bopp, L., Braconnot, P., Brockmann, P., Cadule, P., Cheruy, F., Codron, F., Cozic, A., Cugnet, D., ... Vuichard, N. (2013). Climate change projections using the IPSL-CM5 Earth System Model: From CMIP3 to CMIP5. *Climate Dynamics*, 40(9–10), 2123–2165. <https://doi.org/10.1007/s00382-012-1636-1>
- Dunne, J. P., John, J. G., Adcroft, A. J., Griffies, S. M., Hallberg, R. W., Shevliakova, E., Stouffer, R. J., Cooke, W., Dunne, K. A., Harrison, M. J., Krasting, J. P., Malyshev, S. L., Milly, P. C. D., Phillipps, P. J., Sentman, L. T., Samuels, B. L., Spelman, M. J., Winton, M., Wittenberg, A. T., & Zadeh, N. (2012). GFDL's ESM2 Global Coupled Climate–Carbon Earth System Models. Part I: Physical Formulation and Baseline Simulation Characteristics. *Journal of Climate*, 25(19), 6646–6665. <https://doi.org/10.1175/JCLI-D-11-00560.1>
- Escriva-Bou, A., Mccann, H., Hanak, E., Lund, J., Gray, B., Blanco, E., Jezdimirovic, J., Magnuson-Skeels, B., & Tweet, A. (2020). Water Accounting in Western US, Australia, and Spain: Comparative Analysis. *Journal of Water Resources Planning and Management*, 146(3), 04020004–04020004. [https://doi.org/10.1061/\(ASCE\)WR.1943-5452.0001157](https://doi.org/10.1061/(ASCE)WR.1943-5452.0001157)
- Fernandez-Bou, A. S., Ortiz-Partida, J. P., Dobbin, K. B., Flores-Landeros, H., Bernacchi, L. A., & Medellín-Azuara, J. (2021). Underrepresented, understudied, underserved: Gaps and opportunities for advancing justice in disadvantaged communities. *Environmental Science and Policy*, 122(April), 92–100. <https://doi.org/10.1016/j.envsci.2021.04.014>
- Fletcher, S., Hadjimichael, A., Quinn, J., Osman, K., Giuliani, M., Gold, D., Figueroa, A. J., & Gordon, B. (2022). Equity in Water Resources Planning: A Path Forward for Decision Support Modelers.

Journal of Water Resources Planning and Management, 148(7).

[https://doi.org/10.1061/\(ASCE\)WR.1943-5452.0001573](https://doi.org/10.1061/(ASCE)WR.1943-5452.0001573)

Fletcher, S., Lickley, M., & Strzepek, K. (2019). Learning about climate change uncertainty enables flexible water infrastructure planning. *Nature Communications*, 10(1), 1–11.

<https://doi.org/10.1038/s41467-019-09677-x>

Fletcher, S., Miotti, M., Swaminathan, J., Klemun, M., Strzepek, K., & Siddiqi, A. (2017). Water supply infrastructure planning decision-making framework to classify multiple uncertainties and evaluate flexible design. *Journal of Water Resources Planning and Management*, 143(10), 04017061–04017061.

[https://doi.org/10.1061/\(ASCE\)WR.1943-5452.0000823](https://doi.org/10.1061/(ASCE)WR.1943-5452.0000823)

Gazzotti, P., Emmerling, J., Marangoni, G., Castelletti, A., Wijnst, K. V. D., Hof, A., & Tavoni, M. (2021). Persistent inequality in economically optimal climate policies. *Nature Communications*, 1–10.

<https://doi.org/10.1038/s41467-021-23613-y>

Gent, P. R., Danabasoglu, G., Donner, L. J., Holland, M. M., Hunke, E. C., Jayne, S. R., Lawrence, D. M., Neale, R. B., Rasch, P. J., Vertenstein, M., Worley, P. H., Yang, Z.-L., & Zhang, M. (2011). The Community Climate System Model Version 4. *Journal of Climate*, 24(19), 4973–4991.

<https://doi.org/10.1175/2011JCLI4083.1>

Giglio, R. J., & Wrightington, R. (1972). Methods for apportioning costs among participants in regional systems. *Water Resources Research*, 8(5), 1133–1144.

Giuliani, M., Castelletti, A., Pianosi, F., Mason, E., & Reed, P. M. (2016). Curses, tradeoffs, and scalable management: Advancing evolutionary multiobjective direct policy search to improve water reservoir operations. *Journal of Water Resources Planning and Management*, 142(2), 04015050–04015050.

[https://doi.org/10.1061/\(ASCE\)WR.1943-5452.0000570](https://doi.org/10.1061/(ASCE)WR.1943-5452.0000570)

- Giuliani, M., Lamontagne, J. R., Reed, P. M., & Castelletti, A. (2021). A State-of-the-Art Review of Optimal Reservoir Control for Managing Conflicting Demands in a Changing World. *Water Resources Research*, *57*(12). <https://doi.org/10.1029/2021WR029927>
- Gold, D. F., Reed, P. M., Gorelick, D. E., & Characklis, G. W. (2022). Power and Pathways: Exploring Robustness, Cooperative Stability, and Power Relationships in Regional Infrastructure Investment and Water Supply Management Portfolio Pathways. *Earth's Future*, *10*(2). <https://doi.org/10.1029/2021ef002472>
- Gorelick, D. E., Gold, D. F., Asefa, T., Svrclin, S., Wang, H., Wanakule, N., Reed, P. M., & Characklis, G. W. (2023). Water Supply Infrastructure Investments Require Adaptive Financial Assessment: Evaluation of Coupled Financial and Water Supply Dynamics. *Journal of Water Resources Planning and Management*, *149*(3). <https://doi.org/10.1061/JWRMD5.WRENG-5863>
- Griffies, S. M., Winton, M., Donner, L. J., Horowitz, L. W., Downes, S. M., Farneti, R., Gnanadesikan, A., Hurlin, W. J., Lee, H.-C., Liang, Z., Palter, J. B., Samuels, B. L., Wittenberg, A. T., Wyman, B. L., Yin, J., & Zadeh, N. (2011). The GFDL CM3 Coupled Climate Model: Characteristics of the Ocean and Sea Ice Simulations. *Journal of Climate*, *24*(13), 3520–3544. <https://doi.org/10.1175/2011JCLI3964.1>
- Groves, D., Fischbach, J., Kalra, N., Molina-Perez, E., Yates, D., Purkey, D., Fencel, A., Mehta, V., Wright, B., & Pyke, G. (2014). *Developing Robust Strategies for Climate Change and Other Risks: A Water Utility Framework*. Water Research Foundation. <https://doi.org/10.7249/RR977>
- Haasnoot, M., Kwakkel, J. H., Walker, W. E., & ter Maat, J. (2013). Dynamic adaptive policy pathways: A method for crafting robust decisions for a deeply uncertain world. *Global Environmental Change*. <https://doi.org/10.1016/j.gloenvcha.2012.12.006>
- Hagerty, N. (2023). *Liquid Constrained in California: Estimating the Potential Gains from Water Markets*. https://hagertynw.github.io/webfiles/Liquid_Constrained_in_California.pdf

- Hamilton, A. L., Zeff, H. B., Characklis, G. W., & Reed, P. M. (2022). Resilient California water portfolios require infrastructure investment partnerships that are viable for all partners. *Earth's Future*, 10(4), e2021EF002573–e2021EF002573. <https://doi.org/10.1029/2021ef002573>
- Hanak, E., Jezdimirovic, J., Escriva-Bou, A., & Ayres, A. (2020). *A Review of Groundwater Sustainability Plans in the San Joaquin Valley (Public comments submitted to the California Department of Water Resources)* (pp. 1–13). <https://www.ppic.org/wp-content/uploads/ppic-review-of-groundwater-sustainability-plans-in-the-san-joaquin-valley.pdf>
- Hanak, E., Jezdimirovic, J., Green, S., Escriva-Bou, A., Bostic, D., & Mccann, H. (2018). *Replenishing Groundwater in the San Joaquin Valley*. <https://www.ppic.org/wp-content/uploads/r-0417ehr.pdf>
- Hanak, E., Lund, J., Arnold, B., Escriva-Bou, A., Gray, B., Green, S., Harter, T., Howitt, R., Macewan, D., Medellín-Azuara, J., Moyle, P., & Seavy, N. (2017). *Water Stress and a Changing San Joaquin Valley*. https://www.ppic.org/content/pubs/report/R_0317EHR.pdf
- Hansen, K., Mullin, M., & Riggs, E. K. (2020). Collaboration Risk and the Choice to Consolidate Local Government Services. *Perspectives on Public Management and Governance*, 3(3), 223–238. <https://doi.org/10.1093/ppmgov/gvz017>
- Herman, J. D., Quinn, J. D., Steinschneider, S., Giuliani, M., & Fletcher, S. (2020). Climate adaptation as a control problem: Review and perspectives on dynamic water resources planning under uncertainty. *Water Resources Research*, 56, e24389–e24389. <https://doi.org/10.1029/2019wr025502>
- Herman, J. D., Zeff, H. B., Reed, P. M., & Characklis, G. W. (2014). Beyond optimality: Multistakeholder robustness tradeoffs for regional water portfolio planning under deep uncertainty. *Water Resources Research*, 50(10), 7692–7713. <https://doi.org/10.1002/2014WR015338>

- Jafino, B. A., Kwakkel, J. H., & Taebi, B. (2021). Enabling assessment of distributive justice through models for climate change planning: A review of recent advances and a research agenda. *Wiley Interdisciplinary Reviews: Climate Change*, e721–e721. <https://doi.org/10.1002/wcc.721>
- Jain, S., Scaife, A. A., Shepherd, T. G., Deser, C., Dunstone, N., Schmidt, G. A., Trenberth, K. E., & Turkington, T. (2023). Importance of internal variability for climate model assessment. *Npj Climate and Atmospheric Science*, 6(1), 68. <https://doi.org/10.1038/s41612-023-00389-0>
- Kasprzyk, J. R., Nataraj, S., Reed, P. M., & Lempert, R. J. (2013). Many objective robust decision making for complex environmental systems undergoing change. *Environmental Modelling and Software*, 42, 55–71. <https://doi.org/10.1016/j.envsoft.2012.12.007>
- Kocis, T. N., & Dahlke, H. E. (2017). Availability of high-magnitude streamflow for groundwater banking in the Central Valley, California. *Environmental Research Letters*, 12(8), 084009–084009. <https://doi.org/10.1088/1748-9326/aa7b1b>
- Lehner, F., & Deser, C. (2023). Origin, importance, and predictive limits of internal climate variability. *Environmental Research: Climate*, 2(2), 023001. <https://doi.org/10.1088/2752-5295/accf30>
- Lehner, F., Deser, C., Maher, N., Marotzke, J., Fischer, E., Brunner, L., Knutti, R., & Hawkins, E. (2020). Partitioning climate projection uncertainty with multiple Large Ensembles and CMIP5/6. *Earth System Dynamics*, 11, 491–508. <https://doi.org/10.5194/esd-2019-93>
- Lempert, R. J. (2002). A new decision sciences for complex systems. *Proceedings of the National Academy of Sciences of the United States of America*, 99(SUPPL. 3), 7309–7313. <https://doi.org/10.1073/pnas.082081699>
- Liang, X., Lettenmaier, D. P., Wood, E. F., & Burges, S. J. (1994). A simple hydrologically based model of land surface water and energy fluxes for general circulation models. *Journal of Geophysical Research: Atmospheres*, 99(D7), 14415–14428. <https://doi.org/10.1029/94JD00483>

- Madani, K. (2010). Game theory and water resources. *Journal of Hydrology*.
<https://doi.org/10.1016/j.jhydrol.2009.11.045>
- Martin, G. M., Bellouin, N., Collins, W. J., Culverwell, I. D., Halloran, P. R., Hardiman, S. C., Hinton, T. J., Jones, C. D., McDonald, R. E., McLaren, A. J., O'Connor, F. M., Roberts, M. J., Rodriguez, J. M., Woodward, S., Best, M. J., Brooks, M. E., Brown, A. R., Butchart, N., Dearden, C., ... Wiltshire, A. (2011). The HadGEM2 family of Met Office Unified Model climate configurations. *Geoscientific Model Development*, 4(3), 723–757. <https://doi.org/10.5194/gmd-4-723-2011>
- McPhail, C., Maier, H. R., Kwakkel, J. H., Giuliani, M., Castelletti, A., & Westra, S. (2018). Robustness Metrics: How Are They Calculated, When Should They Be Used and Why Do They Give Different Results? *Earth's Future*, 6(2), 169–191. <https://doi.org/10.1002/2017EF000649>
- Moallemi, E. A., de Haan, F. J., Hadjikakou, M., Khatami, S., Malekpour, S., Smajgl, A., Smith, M. S., Voinov, A., Bandari, R., Lamichhane, P., Miller, K. K., Nicholson, E., Novalia, W., Ritchie, E. G., Rojas, A. M., Shaikh, M. A., Szetey, K., & Bryan, B. A. (2021). Evaluating Participatory Modeling Methods for Co-creating Pathways to Sustainability. *Earth's Future*, 9(3).
<https://doi.org/10.1029/2020EF001843>
- Moallemi, E. A., Kwakkel, J., de Haan, F. J., & Bryan, B. A. (2020). Exploratory modeling for analyzing coupled human-natural systems under uncertainty. *Global Environmental Change*, 65, 102186–102186. <https://doi.org/10.1016/j.gloenvcha.2020.102186>
- Molina-Perez, E., Groves, D., Popper, S., Ramirez, A., & Crespo-Elizondo, R. (2019). *Developing a Robust Water Strategy for Monterrey, Mexico: Diversification and Adaptation for Coping with Climate, Economic, and Technological Uncertainties*. RAND Corporation. <https://doi.org/10.7249/RR3017>
- Osman, K. K., & Faust, K. M. (2021). Toward Operationalizing Equity in Water Infrastructure Services: Developing a Definition of Water Equity. *ACS ES&T Water*, 1(8), 1849–1858.
<https://doi.org/10.1021/acsestwater.1c00125>

- Quinn, J. D., Reed, P. M., Giuliani, M., & Castelletti, A. (2017). Rival framings: A framework for discovering how problem formulation uncertainties shape risk management trade-offs in water resources systems. *Water Resources Research*, 53(8), 7208–7233. <https://doi.org/10.1002/2017WR020524>
- Smith, R., Kasprzyk, J., & Dilling, L. (2019). Testing the potential of Multiobjective Evolutionary Algorithms (MOEAs) with Colorado water managers. *Environmental Modelling and Software*, 117, 149–163. <https://doi.org/10.1016/j.envsoft.2019.03.011>
- Smith, R., Zagona, E., Kasprzyk, J., Bonham, N., Alexander, E., Butler, A., Prairie, J., & Jerla, C. (2022). Decision Science Can Help Address the Challenges of Long-Term Planning in the Colorado River Basin. *JAWRA Journal of the American Water Resources Association*. <https://doi.org/10.1111/1752-1688.12985>
- Srikrishnan, V., Lafferty, D. C., Wong, T. E., Lamontagne, J. R., Quinn, J. D., Sharma, S., Molla, N. J., Herman, J. D., Sriver, R. L., Morris, J. F., & Lee, B. S. (2022). Uncertainty Analysis in Multi-Sector Systems: Considerations for Risk Analysis, Projection, and Planning for Complex Systems. *Earth's Future*, 10(8). <https://doi.org/10.1029/2021EF002644>
- Statutory Water Rights Law and Related California Code Sections (2017). <https://h8b186.p3cdn2.secureserver.net/wp-content/uploads/2017/06/wrlaws.pdf>
- Szeptycki, Leon F., Forgie, Julia, Hook, Elizabeth, Lorick, Kori, & Womble, Philip. (2015). *Environmental water rights transfers: A review of state laws*. Water in the West. <https://waterinthewest.stanford.edu/sites/default/files/WITW-WaterRightsLawReview-2015-FINAL.pdf>
- Taylor, K. E., Stouffer, R. J., & Meehl, G. A. (2012). An Overview of CMIP5 and the Experiment Design. *Bulletin of the American Meteorological Society*, 93(4), 485–498. <https://doi.org/10.1175/BAMS-D-11-00094.1>

- Trindade, B. C., Gold, D. F., Reed, P. M., Zeff, H. B., & Characklis, G. W. (2020). Water pathways: An open source stochastic simulation system for integrated water supply portfolio management and infrastructure investment planning. *Environmental Modelling and Software*, 132. <https://doi.org/10.1016/j.envsoft.2020.104772>
- U.S. Bureau of Reclamation. (2020). *Friant-Kern Canal Middle Reach Capacity Correction Project*. <https://www.usbr.gov/mp/docs/fkc-feasibility-report.pdf>
- Voldoire, A., Sanchez-Gomez, E., Salas Y Mélia, D., Decharme, B., Cassou, C., Sénési, S., Valcke, S., Beau, I., Alias, A., Chevallier, M., Déqué, M., Deshayes, J., Douville, H., Fernandez, E., Madec, G., Maisonnave, E., Moine, M.-P., Planton, S., Saint-Martin, D., ... Chauvin, F. (2013). The CNRM-CM5.1 global climate model: Description and basic evaluation. *Climate Dynamics*, 40(9–10), 2091–2121. <https://doi.org/10.1007/s00382-011-1259-y>
- Volodin, E. M., Dianskii, N. A., & Gusev, A. V. (2010). Simulating present-day climate with the INMCM4.0 coupled model of the atmospheric and oceanic general circulations. *Izvestiya, Atmospheric and Oceanic Physics*, 46(4), 414–431. <https://doi.org/10.1134/S000143381004002X>
- Watanabe, M., Suzuki, T., O'ishi, R., Komuro, Y., Watanabe, S., Emori, S., Takemura, T., Chikira, M., Ogura, T., Sekiguchi, M., Takata, K., Yamazaki, D., Yokohata, T., Nozawa, T., Hasumi, H., Tatebe, H., & Kimoto, M. (2010). Improved Climate Simulation by MIROC5: Mean States, Variability, and Climate Sensitivity. *Journal of Climate*, 23(23), 6312–6335. <https://doi.org/10.1175/2010JCLI3679.1>
- Watkins, A. R. (1998). Cost Allocation in Urban Infrastructure Funding. *Journal of Urban Planning and Development*, 124(1), 44–53. [https://doi.org/10.1061/\(ASCE\)0733-9488\(1998\)124:1\(44\)](https://doi.org/10.1061/(ASCE)0733-9488(1998)124:1(44))
- Womble, P., & Hanemann, W. M. (2020). Water Markets, Water Courts, and Transaction Costs in Colorado. *Water Resources Research*, 56(4), e2019WR025507. <https://doi.org/10.1029/2019WR025507>

Woodruff, M. J., Reed, P. M., & Simpson, T. W. (2013). Many objective visual analytics: Rethinking the design of complex engineered systems. *Structural and Multidisciplinary Optimization*, 48, 201–219. <https://doi.org/10.1007/s00158-013-0891-z>

Yoon, J., Romero-Lankao, P., Yang, Y. C. E., Klassert, C., Urban, N., Kaiser, K., Keller, K., Yarlagadda, B., Voisin, N., Reed, P. M., & Moss, R. (2022). A Typology for Characterizing Human Action in MultiSector Dynamics Models. *Earth's Future*, 10(8). <https://doi.org/10.1029/2021EF002641>

Zeff, H. B., Hamilton, A. L., Malek, K., Herman, J. D., Cohen, J. S., Medellin-Azuara, J., Reed, P. M., & Characklis, G. W. (2021). California's food-energy-water system: An open source simulation model of adaptive surface and groundwater management in the Central Valley. *Environmental Modelling and Software*, 141, 105052–105052. <https://doi.org/10.1016/j.envsoft.2021.105052>

Reviewers' Comments:

Reviewer #1:

Remarks to the Author:

Thank you for your efforts to address all the comments and revise the paper substantially. I think it's an excellent paper which I will use in future research.

Reviewer #2:

Remarks to the Author:

This is a robust and innovative article.

I thank the authors for their detail response to my comments.

I believe the latest version of the manuscript addresses my comments in full.

I therefore recommend the paper for publication.

Reviewer #3:

Remarks to the Author:

I applaud the authors for their commitment to scientific integrity in finding the error in the model and rerunning the entire experiment to remedy it.

The authors did an exceptional job in responding to all the reviewers' comments, and I can safely say that they have addressed my concerns. I look forward to seeing this work published and any follow-on work that comes out of this study.

Response to Reviewers for submission to Nature Communications: "Resilient water infrastructure partnerships in institutionally complex systems face challenging supply and financial risk tradeoffs"

We would like to thank the editor and three anonymous reviewers for taking the time and effort to review our paper for a second time. We are pleased to hear that all three reviewers are satisfied with the previous revision and response. The previous set of reviews were very constructive and helped to improve the quality and impact of our work.

Reviewer comment

The only remaining comment was from Reviewer #2:

"I opened the code repository. Although the repository is very comprehensive, I think a READ me file is missing. I recommend to the authors to include such a file such that interested readers are able to navigate more efficiently through the directory structure and attempt to replicate the results in the paper."

Response to reviewer comment

We commend Reviewer #2 for taking the time and effort to review the repository for this work. We would like to note that our repository already did contain a README file with a comprehensive set of instructions for reproducing the optimization, analysis, and figures. However, we acknowledge that perhaps this file was not displayed prominently enough within the software repository to be seen by all reviewers and future users.

To improve the transparency and reproducibility of our work in line with the reviewer's comment, we have taken several steps:

1. First, for the new updated code associated with this final revised manuscript submission, we followed the same three steps as were followed with previous submissions:
 - a. All code and data for this work are contained within a particular branch of the CALFEWS model repository (https://github.com/hbz5000/CALFEWS/tree/MORDM_experiment_paper1).
 - b. The lead author Andrew Hamilton created a tagged release of this branch on his own fork of the repository – this release contains zip file archive of all code and data used to produce the paper, including all model results data used to create the figures (<https://github.com/ahamilton144/CALFEWS/releases/tag/v2.1>).
 - c. A permanent archived copy of this release was also created in Zenodo, including a permanent identifier DOI (<https://doi.org/10.5281/zenodo.12789172>).
2. Based on the reviewer's comment, we added references to the README file and its instructions more clearly into the descriptions of the GitHub release and the permanent Zenodo archive. Now a prospective user will know exactly where to locate these instructions without having to search through the zipped archives.
3. Additionally, following emerging best practices for scientific research, we created a Metarepository for this paper (https://github.com/IMMM-SFA/hamilton-et al_2024_naturecommunications).
 - a. A metarepository is a single point of access to find instructions and links to software and datasets used in a scientific work. This is based on a metarepository GitHub template

developed by Department of Energy researchers (<https://github.com/IMMM-SFA/metarepo>).

- b. The metarepository does not actually contain the code and data used in this work; rather, it provides an overview of sources used in this work and links directly to the GitHub release and Zenodo archive outlined in Step 1 above.
- c. To further improve clarity based on the reviewer comment, we included instructions in the metarepository README file that direct the user to the corresponding README file in the underlying CALFEWS software release. The metarepository provide a higher-level set of instructions and direct the user to the underlying CALFEWS software release for more detailed instructions on reproducing the experiment.
- d. Following best practices in FAIR science, we then created a tagged GitHub release and a permanent Zenodo archive for the metarepository itself (<https://doi.org/10.5281/zenodo.12801237>). These are the resources that are now cited in the manuscript.